# Looking at the Pathogenesis of the Rabies Lyssavirus Strain Pasteur Vaccins through a Prism of the Disorder-Based Bioinformatics

**DOI:** 10.3390/biom12101436

**Published:** 2022-10-07

**Authors:** Surya Dhulipala, Vladimir N. Uversky

**Affiliations:** 1Department of Molecular Medicine, Morsani College of Medicine, University of South Florida, Tampa, FL 33612, USA; 2USF Health Byrd Alzheimer’s Research Institute, Morsani College of Medicine, University of South Florida, Tampa, FL 33612, USA; 3Protein Research Group, Institute for Biological Instrumentation of the Russian Academy of Sciences, Federal Research Center “Pushchino Scientific Center for Biological Research of the Russian Academy of Sciences”, 142290 Pushchino, Moscow Region, Russia

**Keywords:** rabies, intrinsic disorder, intrinsically disordered protein, intrinsically disordered protein region, protein–protein interaction

## Abstract

Rabies is a neurological disease that causes between 40,000 and 70,000 deaths every year. Once a rabies patient has become symptomatic, there is no effective treatment for the illness, and in unvaccinated individuals, the case-fatality rate of rabies is close to 100%. French scientists Louis Pasteur and Émile Roux developed the first vaccine for rabies in 1885. If administered before the virus reaches the brain, the modern rabies vaccine imparts long-lasting immunity to the virus and saves more than 250,000 people every year. However, the rabies virus can suppress the host’s immune response once it has entered the cells of the brain, making death likely. This study aimed to make use of disorder-based proteomics and bioinformatics to determine the potential impact that intrinsically disordered protein regions (IDPRs) in the proteome of the rabies virus might have on the infectivity and lethality of the disease. This study used the proteome of the Rabies lyssavirus (RABV) strain Pasteur Vaccins (PV), one of the best-understood strains due to its use in the first rabies vaccine, as a model. The data reported in this study are in line with the hypothesis that high levels of intrinsic disorder in the phosphoprotein (P-protein) and nucleoprotein (N-protein) allow them to participate in the creation of Negri bodies and might help this virus to suppress the antiviral immune response in the host cells. Additionally, the study suggests that there could be a link between disorder in the matrix (M) protein and the modulation of viral transcription. The disordered regions in the M-protein might have a possible role in initiating viral budding within the cell. Furthermore, we checked the prevalence of functional disorder in a set of 37 host proteins directly involved in the interaction with the RABV proteins. The hope is that these new insights will aid in the development of treatments for rabies that are effective after infection.

## 1. Introduction

Rabies lyssavirus is a bullet-shaped, negative-sense, unsegmented, single-stranded RNA virus of the *Rhabdoviridae* family. There are 10 viruses in the Rabies serogroup, but most are not pathogenic to humans. Rabies lyssavirus and Australian bat lyssavirus are the only two rhabdoviruses that have been known to cause disease in humans [1]. The rabies virus (RABV) is a zoonotic neurotropic virus that causes fatal neurological symptoms in almost all mammals and is spread through the bite of an infected mammal. Rabies disease causes between 40,000 and 70,000 deaths every year worldwide. Once a rabies patient has become symptomatic, there is no effective treatment for the illness. In fact, in unvaccinated individuals, the case-fatality rate of rabies is close to 100% [2]. French scientists Louis Pasteur and Émile Roux developed the first vaccine for rabies in 1885. If administered before the virus reaches the brain, the modern rabies vaccine imparts long-lasting immunity to the virus. It saves more than 250,000 people every year. RABV is able to suppress the host’s immune response once it has entered the cells of the brain, making death likely if the vaccine is not administered [3]. The virus works by infecting muscle cells through saliva in bites before traveling to the brain. The virus travels through neuromuscular junctions and neural pathways using axoplasmic transport. Once the virus reaches the brain, it triggers endocytosis into neurons. It then undergoes transcription and replication by polymerase shuttering [4]. The virus suppresses the antiviral response and infects more cells before eventually being released and transported to the salivary glands. The high neuroinvasiveness of RABV has been attributed to the ability of the virus to evade immune responses through various means and to conserve the structures of neurons [3].

All RABVs encode for five major proteins, as well as four more isoforms of the phosphoprotein that are generated through alternative initiation. The major proteins are the phosphoprotein (P-protein), matrix protein (M-protein), glycoprotein (G-protein), nucleoprotein (N-protein), and polymerase, or the large protein (L-protein). A total of five phosphoproteins can be made through alternative initiation [5]. The RNA genome of the virus is roughly 11 kb in size. A graphical representation of the virus can be seen in Figure 1. The P-protein is bound tightly around the RNA of the virus to create the RNP (ribonucleoprotein) core, and the RNP condenses with the N- and L-proteins to create the helical nucleocapsid of the virus. The N-protein homomultimerizes to form the nucleocapsid and binds with the P-protein [6]. The P-protein forms a homotrimer when phosphorylated and bound to the L-protein, which positions the L-protein over the template RNA strands [7]. The M-protein forms a homomultimer between the nucleocapsid core and the glycoprotein spikes that surround the virus to connect the two. The G-protein spikes, which are homotrimers of the G-protein, form the sole ligand for the cellular receptor. Anti-rabies antibodies target the G-protein homotrimers [8,9].

Once the virus reaches the cell, the G-protein initiates endocytosis. The low pH inside the endosome initiates the fusion of the membrane with the endosome, resulting in the release of the nucleocapsid into the cell. The P-protein acts as a cofactor to the L-protein, creating the functional viral polymerase [11]. The M-protein acts as a bridge between the plasma membrane and the generated nucleocapsids, providing a crucial role in virus budding [3,12,13]. The condensing of the nucleocapsid and the final shape is mediated by the M-protein [3].

The P-protein has an important role in the infection and propagation of RABV. It binds to the dynein light-chain proteins DYNLL1 and DYNLL2 in humans, allowing the virus to be transported along the axon to the nucleus. During transcription, alternative initiation using a ribosomal leaky scanning mechanism can create five variations of the P-protein, termed P1–5 [14]. This enables higher flexibility in the action of the P-protein. It interacts with the ribosomal protein L9 during the initial stages of RABV infection to inhibit RABV transcription during this period, controlling viral replication [15]. The P-protein also acts as an antagonist to STAT1 and STAT2 proteins within the cell. STAT proteins, which are activated by type-I-interferon receptors in the type-I-interferon-mediated innate immune response, are cytoplasmic signal transducers and transcription activators associated with the immune response [16,17]. The P-protein exploits microtubule processes within the cell to transport the virus and force STAT1 to use microtubule-inhibited mechanisms, suppressing the nuclear transport of STAT and signaling [18]. STAT proteins are important for the establishment of the antiviral state in the cell. The STAT-targeting feature of interferon antagonists is considered a determining factor in the pathogenicity of the virus. Inhibition of the interaction between the P-protein and STAT has been shown to severely reduce lethality and increase the immune response to RABV. The N-protein forms a complex with the P-protein to encapsulate the RNA genome of the virus and protect it from nuclease activity and phosphorylation [6]. The N-protein has been shown to help suppress the type-3-interferon-mediated immune system by evading the RIG-I-mediated antiviral response [19,20]. The L-protein, due to its activity as a viral polymerase, plays multiple enzymatic roles in the synthesis and processing of viral RNA [21]. It acts as a low-fidelity polymerase, resulting in a high rate of mutations in RABV.

The M-protein has been shown to be significant in regulating viral budding. It has been shown to bind to the plasma membrane even in the absence of other viral proteins, which suggests that the protein acts to connect the plasma membrane to the nucleocapsid. The M-protein has been shown to be able to induce vesicle budding without necessitating interactions with other proteins [3,12]. The infection of a cell with an M-protein-deficient RABV mutant was observed to result in smaller numbers of rod-shaped or round viral particles rather than the bullet-shaped viruses that are standard with mature RABV [13]. The M-protein has been shown to inhibit viral transcription in many viruses, including RABV, suggesting that it has a highly regulatory role in transcription and replication [12,22]. The M-protein has been shown to interact with RelAp43, a protein in the NF-*κ*B family of proteins, which results in the further suppression of the antiviral immune response [23]. The G-protein is associated with the acceleration of the budding efficiency of the virus [12,13]. The G-protein also drastically increases pathogenicity because, as the only surface protein in the virus, it induces innate immune responses by binding to immune receptors [24]. This binding is shown to promote effective virus uptake, which drastically increases virulence.

As is typical for other viruses, all RABV proteins are multifunctional. Such multifunctionality is crucial, as it allows a very small set of viral proteins to manage and control each and any aspect of the viral “life”, from entry to replication to the formation and exit of new infectious particles, and regulate each and any aspect of virus interaction with the host. In fact, a typical protein repertoire of a typical virus includes a minimal set of specific structural proteins that are crucial for the viral particle assembly and a set of non-structural proteins needed for the hijacking of many functional pathways of the host cell. This is why many viral proteins are multifunctional. They are typically engaged in numerous interactions with various host cell components. In the protein universe, multifunctionality and binding promiscuity are typically associated with protein intrinsic disorder. This directly follows from the recognition that many protein functions do not need a unique 3D structure [25,26,27,28,29,30,31,32,33] and that such structure-less intrinsically disordered proteins (IDPs) and intrinsically disordered protein regions (IDPRs) are commonly found in various proteomes [34,35,36,37,38], where they are involved in regulation, signaling, and control pathways [27,30,32,39,40,41], thereby possessing functions that complement the functional repertoire of traditional ordered proteins [42,43,44,45,46,47]. IDPs/IDPRs are often involved in various human diseases [48,49]. They have complex mosaic structures and show remarkable multi-level spatiotemporal heterogeneity, existing as dynamic conformational ensembles [25,27,31,40,42,50,51] where different parts of a protein can be (dis)ordered to different degrees [40,52,53]. Importantly, these differently (dis)ordered pieces of the protein structural mosaic might have well-defined and specific functions [53]. Therefore, IDPs/IDPRs are structurally and functionally heterogeneous complex systems whose functionality is described in terms of the protein structure–function continuum [53,54], where the structural and functional diversification of a protein is defined by several factors determining the capability of a single gene to encode a set of distinct protein molecules, known as proteoforms [55]. This is achieved at several levels by altering the chemical structure of the proteinaceous product(s) of a given gene via allelic variations at the DNA level (utilizing several specific means, such as single or multiple point mutations, indels, and SNPs) by alternative splicing and other pre-translational mechanisms affecting mRNA, via a broad arsenal of countless post-translational modifications (PTMs) of a polypeptide chain, by the presence of intrinsic disorder, or by structural alterations induced by functioning [54].

Importantly, based on computational analyses of the abundance of intrinsic disorder in various organisms, it has been concluded that the proteomes of viruses have the largest variability in the content of disordered residues in comparison with all other kingdoms of life [37,38]. The abundance and functional importance of IDPs/IDPRs have been systematically investigated for human papillomaviruses (HPVs) [56,57], human immunodeficiency virus 1 (HIV-1) [58], human hepatitis C virus (HCV) [59,60], Dengue virus [61], rotavirus [62], human respiratory syncytial virus [63], Zika virus [64,65], Chikungunya virus [66], Alkhurma virus (ALKV) [67], Japanese encephalitis virus [68], and SARS-CoV-2, human SARS, and bat SARS-like coronaviruses [69]. These studies suggested that the presence of IDPRs in viral proteins is crucial for their functionality and represents an important means of the overall enhancement of viral propagation during the virus life cycle. To the best of our knowledge, there is no similar analysis of the disorder status of the RABV proteome. The aim of this study was to fill this gap by conducting a comprehensive computational analysis of the penetrance of intrinsic disorder in the proteins of the Rabies lyssavirus strain Pasteur Vaccins proteome.

## 2. Materials and Methods

The Universal Protein Resource (UniProt) is an annotated database of protein sequences [70]. Entries may be manually annotated with information extracted from the literature and evaluated by computational analysis (Swiss-Prot) or computationally analyzed (TrEMBL). The Rabies lyssavirus strain Pasteur Vaccins proteome is a Swiss-Prot annotated entry. Each entry contains information, including but not limited to the taxonomy, molecular function and included biological processes, subcellular location, potential modifications, pathology, interactions, and structures. Of these, the amino acid sequences and the structures are the most valuable resources for the purposes of studying intrinsic disorder in the virus. A set of 37 human proteins involved in interactions with the RABV proteins was assembled through a literature search, with the majority of data being retrieved from [71]. Amino acid sequences and basic disorder-related features of these proteins are provided in Appendix A.

The amino acid sequences can be analyzed through various methods to identify regions in the protein that have a predisposition towards intrinsic disorder. Through a comparison of these regions of intrinsic disorder and the function of the protein, an analysis can be made of the disorder-based functionality of the protein. Determining the structure of the protein is important for mapping the intrinsic regions of disorder to regions on the protein. Intrinsically disordered regions are flexible. As a result, these regions are not recorded in the crystal structure of the protein. Regions of the protein that do not show up in the crystal structure are indicative of intrinsic disorder [72].

For this study, amino acid FASTA sequences for the Rabies lyssavirus PV proteome were gathered through UniProt. The sequences were run through a series of disorder prediction tools to generate an estimate of the intrinsic disorder of each residue. These predictions of intrinsic disorder were averaged to form an overall prediction of intrinsic disorder for that residue. Table 1 shows a summary of these predictions, including the UniProt entry ID of each protein analyzed. The table also displays the protein length and the length of the longest region of disorder within the protein. Overall disorder was calculated based on the incidence of regions of high disorder. The FASTA sequences used in this study are reproduced in Appendix A. The underlined sections of each amino acid sequence correspond to regions of high disorder within the protein. The amino acid sequences presented here were taken from UniProt [70]. Regions of high intrinsic disorder (intrinsic disorder of more than 30%) are shown using bold underlined text. The per-residue intrinsic disorder propensity for each protein is calculated by taking the averages of the PONDR^®^ VL3, PONDR^®^ VLS2, PONDR^®^ VLXT, PONDR^®^ FIT, IUPred_Short, and IUPred_Long predictors.

Predictor of Natural Disordered Regions (PONDR^®^) is meta-prediction software that can analyze an amino acid FASTA sequence on a per-residue basis to predict regions of intrinsic disorder. This study made use of the PONDR^®^VLXT, PONDR^®^VL3, PONDR^®^VLS2, and PONDR^®^ FIT meta-predictors from the PONDR family, as well as IUPred_Short and IUPred_Long [73,74,75].

PONDR^®^VLXT works by applying three different neural networks: one for each terminal end of an intrinsically disordered sequence and one for the internal region of the sequence. Each network uses a specific dataset that contains only the amino acid residues that are present in that region. The result of the predictor is an average of the results of the three networks. Transitions between the prediction networks work by averaging the predictors in a short region of overlap at the boundary between the two. PONDR^®^ VLXT is useful for predicting short regions of disorder but underestimates the occurrence of long disordered regions. PONDR^®^ VL3 works by running the residue through ten neural networks and selecting the final prediction by taking the simple majority vote of the predictions. This meta-predictor is known to be useful for predicting longer regions of intrinsic disorder. PONDR^®^ VSL2 combines neural network predictors for short and long disordered regions. The networks are trained using sequences of specific lengths, and the final prediction is a weighted average of the predictions for each length. Because it combines both short and long disordered regions, it is considered the most accurate predictor of the three [73,74,75].

IUPred works from the assumption that globular and structured proteins have higher numbers of effective inter-residue interactions than disordered proteins do, which means that they have negative free-energy. Structured proteins have lower free-energy estimates compared to disordered proteins. The IUPred meta-predictor is able to use this biophysics-based approach to estimate disorder by calculating the pairwise free-energy of the sequence [74,75]. IUPred Long predicts global structural disorder, or disorder in regions with more than 30 consecutive residues. IUPred Short is useful for predicting short disordered regions, such as the region corresponding to the missing residues in the X-ray structure of a largely globular protein. ANCHOR2 is used to predict context-dependent intrinsic disorder. Context-dependent intrinsic disorder may occur when the binding region of an IDPR is able to interact specifically with a globular protein. When bound, these regions adopt an ordered structure. Context-dependent intrinsic disorder may also occur when the change in disorder is due to a change in the redox state. These regions may change their disorder depending on their localization relative to the cell. For all query proteins, the presence of such context-dependent disordered regions, disorder-based binding regions, and molecular recognition features (MoRFs), i.e., disordered regions that fold upon their interaction with partners, was analyzed by the ANCHOR algorithm [76,77].

A recently designed computational platform, RIDAO (Rapid Intrinsic Disorder Analysis Online), was used to obtain the intrinsic-disorder-related characteristics of the query proteins [78]. This tool aggregates the results from a number of well-known disorder predictors: PONDR^®^ VLXT [79], PONDR^®^ VL3 [80], PONDR^®^ VLS2B [81], PONDR^®^ FIT [73], IUPred2 (Short), and IUPred2 (Long) [74,82]. Furthermore, RIDAO provides the mean disorder profile (MDP), along with the standard errors. It also performs CH-CDF (charge-hydropathy–cumulative distribution function) analysis of the query proteins [83,84,85] and yields data for ΔCH-ΔCDF plots, which enables rapid discrimination between flavors of disorder [86].

The outputs of the per-residue predictors were averaged, and proteins were grouped based on their percentages of predicted intrinsically disordered residues (PPIDR) using accepted classification criteria [87]. Proteins with an average content of intrinsically disordered residues below 10% are considered ordered or mostly ordered. Proteins containing between 10% and 30% predicted disordered residues are considered moderately disordered. Proteins containing more than 30% predicted disordered residues are considered highly disordered [87].

Complementary disorder evaluations, together with important disorder-related functional information, were retrieved from the D^2^P^2^ database (http://d2p2.pro/, accessed on 15 August 2022) [88], which is a database of predicted disorder for a large library of proteins from completely sequenced genomes [88]. The D^2^P^2^ database uses the outputs of IUPred [74,82], PONDR^®^ VLXT [79], PrDOS [89], PONDR^®^ VSL2B [81], PV2 [88], and ESpritz [90]. The visual console of D^2^P^2^ displays 9 colored bars representing the location of disordered regions, as predicted by these different disorder predictors. In the middle of the D^2^P^2^ plots, the blue–green–white bar shows the predicted disorder agreement between the nine disorder predictors (IUPred, PONDR^®^ VLXT, PONDR^®^ VSL2, PrDOS, PV2, and ESpritz), with blue and green parts corresponding to disordered regions by consensus. Above the disorder consensus bar are two lines with colored and numbered bars that show the positions of the predicted (mostly structured) SCOP domains [91,92] using the SUPERFAMILY predictor [93]. The yellow zigzagged bar shows the location of the predicted disorder-based binding sites (MoRF regions) identified by the ANCHOR algorithm [76], whereas differently colored circles at the bottom of the plot show the locations of various PTMs assigned using the outputs of the PhosphoSitePlus platform [94], which is a comprehensive resource for experimentally determined post-translational modifications.

Information on the interactability of human proteins that interact with RABV proteins was retrieved using Search Tool for the Retrieval of Interacting Genes (STRING, http://string-db.org/, accessed on 15 August 2022). STRING generates a network of protein–protein interactions based on predicted and experimentally validated information on the interaction partners of a protein of interest [95]. In the corresponding network, the nodes correspond to proteins, whereas the edges show predicted or known functional associations. Seven types of evidence are used to build the corresponding network, where they are indicated by differently colored lines: a green line represents neighborhood evidence; a red line represents the presence of fusion evidence; a purple line represents experimental evidence; a blue line represents co-occurrence evidence; a light blue line represents database evidence; a yellow line represents text mining evidence; and a black line represents co-expression evidence [95]. In this study, STRING was utilized in three different modes: to create PPI networks centered on individual human proteins, to generate the internal network of protein–protein interactions (PPIs) among the human proteins involved in interactions with the RABV proteins, and to build a PPI network centered on the entire set.

The propensity of the RABV proteins to undergo liquid–liquid phase separation was evaluated by FuzDrop (Fuzzy Droplet Predictor, https://fuzdrop.bio.unipd.it/predictor, accessed on 15 August 2022) [96].

All computer-generated structures of the RABV proteins analyzed in this study were generated using SWISS-MODEL [97] and ExPasy. The 3D structural models of human proteins that interact with the RABV proteins were generated by AlphaFold [98].

## 3. Results and Discussion

### 3.1. Predicted Disorder of the P-Protein and Its Suggested Functional Consequences

The P-protein of RABV, which is a 297-residue-long catalytic polymerase cofactor and regulatory protein that plays an important role in viral transcription and replication, was shown to display a significant amount of intrinsic disorder (see Figure 2). In fact, based on the PONDR^®^ VSL2 outputs, roughly 49% of the protein residues were predicted to be intrinsically disordered (i.e., they have disorder scores exceeding the 0.5 threshold). Furthermore, almost 46% of its residues were predicted to be flexible (i.e., possessing disorder scores ranging from 0.15 to 0.5), indicating that 95% of this protein is expected to be either disordered of structurally flexible. Figure 2A also shows that the P-protein contains two long intrinsically disordered regions, IDD1 (residues 33–89) and IDD2 (residues 133–199), that flank an oligomerization domain, and a short disordered/flexible region (residues 242–252) located within the mostly ordered C-terminal domain (PCTD, residues 201–297). It was indicated that Negri bodies, also known as viral inclusion bodies in the host cytoplasm used for viral replication, are formed via the interaction of the P-protein oligomerization domain, IDD2, and the PCTD [99] with the intrinsically disordered regions of the N-protein, whereas the N-terminal part and IDD1 of the P-protein are dispensable [99,100].

High levels of intrinsic disorder in the RABV P-protein are further evidenced by the analysis of its X-ray crystal structure (PDB ID: 3OA1). In fact, although the 69–297 fragment of this protein was used in the crystallization experiments, the structure was eventually solved for less than half of this construct (residues 192–297), with the entire N-terminal half (residues 69–191) representing a region with missing electron density (i.e., highly flexible or disordered region). Furthermore, even within the solved structure of the C-terminal domain of the P-protein, some short regions were not modeled or incompletely modeled as well (residues 220–221, 231, 272–273, and 296–297) (PDB ID: 3OA1).

Importantly, as per the manually asserted information inferred from the sequence similarity and available in the UniProt database (https://www.uniprot.org/uniprotkb/P06747/entry, accessed on 15 August 2022), most of the IDD2 region of the RABV P-protein is expected to be engaged in binding to cytoplasmic dynein light-chains 1 and 2 (DYNLL1 and DYNLL2, see below). Furthermore, ANCHOR analysis [76,77] suggested that the P-protein contains three potential disorder-based binding sites, known as molecular recognition features (MoRFs), which are disordered regions that are expected to fold upon their interaction with specific partners, thereby driving protein–protein interactions. These are residues 34–90, 124–136, and 166–190. Therefore, it is likely that these segments (shown as gray-shaded areas in Figure 2A) might correspond to ligand binding sites of the RABV P-protein (see below).

Additionally, the high disorder and structural flexibility content of the RABV P-protein suggests that the mechanism of action of this protein to suppress the type-I-interferon-mediated immune response may be based on the utilization of its disordered (or flexible) regions for interaction with the STAT proteins of cells. In fact, although the RABV P-protein binding site responsible for the STAT1/2 interaction is located within the ordered C-terminal domain (CTD, residues 186–297 [16,17,101,102,103,104]), the residues that made the greatest contribution to this interaction and that form the so-called W-hole (C261, W265, and M287) were predicted to be flexible by at least one of the disorder predictors used in this study, with the highest level of structural flexibility being expected for residue M287, which is 100% conserved among most lyssavirus P-proteins [16] and shows a mean disorder score of 0.31 ± 0.12 (ranging from 0.16 to 0.5, as per the outputs of IUPred_short and PONDR^®^ VSL2, respectively (see Figure 2A). Similarly, the positive patch (residues K211, K212, K214, and R260), which is 100% conserved in the lyssavirus P-proteins and known to be responsible for interaction with the N-protein [16], is predicted to be flexible/disordered as well (see Figure 2A). This, again, suggests that structural flexibility plays a role in the interaction of the ordered CTD with partner proteins, including STAT1 and STAT2. It was pointed out that because of these C-terminal-domain-driven interactions of the RABV P-protein (CTD, residues 186–297 [101,102,103,104]) with host STAT proteins, the P-protein represents the major interferon antagonist of the lyssavirus, thereby affecting the type-I-interferon (IFNα/β)-mediated innate immune response [16]. It was also pointed out that the interaction of the RABV P-protein with STATs is crucial for the development of the lethal rabies disease [16].

Another level of structural and functional complexity of this protein is given by the fact at it has multiple isoforms generated by the alternative initiation of the P-protein during viral transcription. In fact, alternative initiation generates isoforms P2, P3, P4, and P5, which differ from the canonical isoform P1 due to a lack of N-terminal residues 1–19, 1–52, 1–68, and 1–82. An obvious consequence of this truncation is the elimination of the first MoRF region of P1, suggesting that these isoforms might be characterized by different interactability. Curiously, although P3, P4, and P5 have all lost an N-terminal MoRF as expected, P2 was predicted to behave differently. In fact, despite missing the N-terminal residues predicted to be the MoRF in P1, this isoform gained three new N-terminal MoRFs (residues 1–14, 19–27, and 30–38).

The functional diversity of the RABV P-protein is further increased by the phosphorylation of its serine residues S63 and S64 by an unknown kinase (denoted rabies virus protein kinase (RVPK)) and residues S162, S210, and S271 by protein kinase C (PKC) [105], all located within IDPRs. A recent study revealed that the phosphorylation of the P3 isoform of the RABC P-protein at the S210 position resulted in a significant reduction in the nuclear localization modulated by the MT binding/bundling of P3 [106].

Therefore, due to the presence of regions with high intrinsic disorder content, several MoRFs, several phosphorylation sites, and the usage of alternative initiation, it is possible for the P-protein to serve many roles within the virus [10]. Furthermore, the P-protein isoforms were shown to differ in nucleocytoplasmic localization and microtubule (MT) association, mediated by several functional motifs, including the nuclear localization sequence (NLS, residues 211–214) and N- and C-terminally located nuclear export sequences (N-NES and C-NES, residues 49–58 and 223–232, respectively) [107,108,109]. For example, shorter isoforms (P3 to P5) lacking the N-terminally located NES are more nuclear and are capable of binding and bundling MTs [107]. As per the outputs of PONDR^®^ VSL2, N-NES and NLS are located within IDPRs (residues 33–89 and 208–216, respectively; see Figure 2A).

Figure 2A shows that intrinsic disorder is unevenly distributed within the P-protein sequence. It is preferentially concentrated at its N-terminal and central regions (residues 1–200), with the C-terminal domain being predicted to possess a more ordered structure. In agreement with the results of the computational evaluation of the intrinsic disorder predisposition of this protein, a crystal structure was solved for residues 192–295 (see Figure 2B) (PDB ID: 3OA1). Curiously, as was already indicated, although a much longer fragment of the P-protein (residues 69–297) was used in the crystallization experiments, a very significant part of this polypeptide was not observed in the resulting structure, representing regions of missing electron density; i.e., regions with high conformational flexibility that preclude them from being crystallized.

Since the P-protein participates in the formation of Negri bodies [99] and can bind PML-bodies [110], we also checked the liquid–liquid phase separation (LLPS) potential of this protein by FuzDrop. This analysis revealed that the longest isoform (P1) of the P-protein is characterized by p_LLPS_ = 0.5276 and contains a droplet-promoting region (DPR, residues 134–184) located within the long central IDPR of this protein (residues 133–199). A recent systematic analysis revealed that this region indeed plays a crucial role in the formation of viral Negri body (NB)-like structures in infected cells [100]. Furthermore, it was shown that the deletion of residues 151–181 did not affect the ability of the RABV P-protein to form NB-like structures, indicating that only the amino-terminal part of IDD2 (residues 132–150) is required for this process [100].

Shorter isoforms of the P-protein, P2, P3, P4, and P5, showed p_LLPS_ values of 0.6561 (DPR, residues 115–166), 0.6832 (DPRs, residues 1–34 and 82–133), 0.4938 (DPRs, residues 1–19 and 66–115), and 0.3526 (DPR, residues 52–103), indicating that alternative initiation dramatically affects the LLPS potential of the P-protein. According to the FuzDrop developers, proteins with p_LLPS_ ≥ 0.60 are droplet-drivers, which can spontaneously undergo LLPS. Droplet-client proteins have pLLPS < 0.60 but possess droplet-promoting regions, which can induce their partitioning into condensates [96]. Therefore, according to the results of this analysis, P2 and P3 isoforms can potentially serve as droplet-drivers, whereas other isoforms (P1, P4, and P5) most likely represent droplet-clients.

FuzDrop also indicated that P1 contains nine regions with context-dependent interactions (residues 20–47, 52–64, 67–85, 94–101, 116–125, 129–138, 188–210, 252–257, and 273–282). These regions can potentially undergo disorder-to-order or disorder-to-disorder transitions but maintain conformational heterogeneity in the bound state and show sensitivity to the cellular context or post-translational modifications, potentially serving as regulatory engines of cellular pathways [111].

### 3.2. Disorder of the M-Protein and Its Suggested Functional Consequences

The M-protein is a 202-residue-long protein that plays a crucial role in the assembly and budding of the virion and engages in complete coverage of the ribonucleoprotein coil to keep it in a condensed bullet-shaped form (see Figure 1). It was found to be highly disordered as well, with 43% of this protein being composed of IDPRs (see Figure 3A). This suggests that the interactions of the M-protein with RelAp43 and other proteins in the NF-*κ*B pathway [23] may induce the suppression of the antiviral response via the utilization of some advantages of intrinsic disorder.

This is in line with the results of several studies on the potential roles of intrinsic disorder in proteins that form the shells of several viruses (such as SARS-CoV-2, MERS-CoV, SARS-CoV, other CoVs, Nipah, Zika, HIV, and retroviruses) for viral transmission, immune evasion, and virulence [112,113,114,115,116,117,118,119,120]. This intrinsic disorder in the M-protein may also allow for the increased flexibility of the protein to aid in the regulation of virus budding. The M-protein has been shown to have the ability to create vesicles without any interaction with other viral proteins, suggesting that the flexibility of the protein allows it to induce budding by itself [3,13]. Figure 3A presents the disorder profile of the MATRX_RABVP protein and shows that a section in the middle of the protein, spanning roughly from residue 50 to residue 175, displays, on average, low disorder content and is likely to represent the structural domain of the protein, which, however, includes some disordered/flexible regions.

Curiously, region 115–151, which is essential for the glycoprotein (as per manually asserted information inferred from the sequence similarity and available in the UniProt database; see https://www.uniprot.org/uniprotkb/P08671/entry, accessed on 15 August 2022) binding, includes an IDPR (residues 129–141, as per PONDR^®^ VLXT, or residues 130–137, as per PONDR^®^ VSL2), indicating that the intrinsic disorder (or structural flexibility) of this region can contribute to its interactability. Furthermore, the M-protein contains a PPxY motif (residues 35–38), which is commonly found in viral proteins capable of manipulating the autophagic machinery to prevent the autophagic degradation of viruses [121]. This PPxY motif is included in the long N-terminal IDPR (residues 1–48) (see Figure 3A).

**Figure 3 biomolecules-12-01436-f003:**
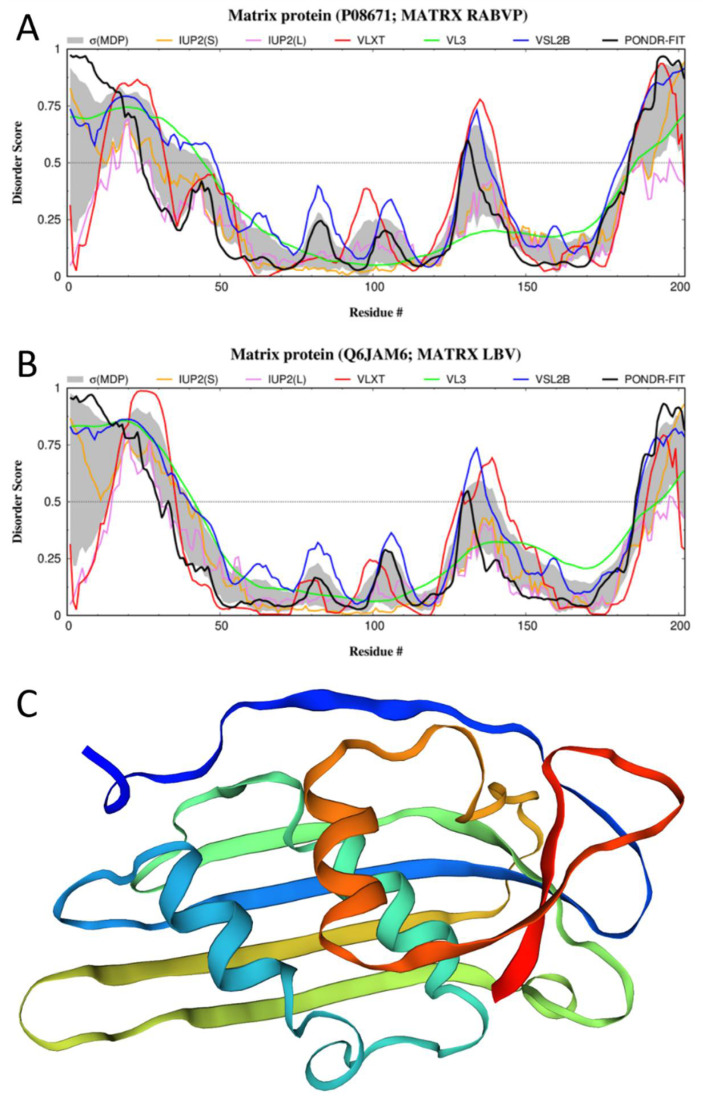
Structure and disorder in the M-protein from RABV (UniProt ID: P08671). (**A**) Intrinsic disorder profile (disorder score vs. residues number (residues #)) generated for the M-protein from the RABV strain VP by RIDAO. (**B**) Intrinsic disorder profile generated for the Lagos bat virus matrix protein (UniProt ID: Q6JAM6) by RIDAO. (**C**) A structural model for the 30–202 fragment of the RABV M-protein that was created by SWISS-MODEL [97] using the structure of the Lagos bat virus matrix protein (PDB ID: 2W2S; [122]) as a template, which shows sequence identity to the query M-protein of 76.73%.

Although no structural information is currently available for the M-protein of RABV, we used SWISS-MODEL (https://swissmodel.expasy.org/, accessed on 15 August 2022) to create homology models for this protein. Figure 3C shows a model for the 30–202 fragment of the RABV M-protein that was created using the structure of the Lagos bat virus matrix protein (PDB ID: 2W2S; [122]; UniProt ID: Q6JAM6) as a template with sequence identity to the query M-protein of 76.73%. A comparison of Figure 3A,B illustrates the remarkable similarity between the per-residue intrinsic disorder predispositions of the Lagos bat virus matrix protein and the RABV M-protein, thereby providing the intrinsic-disorder-based validation of the selection of the Lagos bat virus matrix protein as a template.

Finally, FuzDrop analysis indicated that although the M-protein shows a low probability of spontaneous LLPS (p_LLPS_ = 0.2220), this protein contains one C-terminally located DPR (residues 199–202), which is included in the IDPR (residues 182–202), indicating that the M-protein can act as the droplet-client. FuzDrop also identified regions 16–29, 121–131, and 172–190 as regions with context-dependent interactions [111].

### 3.3. Disorder of the N-Protein and Its Suggested Functional Consequences

The 450-residue-long N-protein is a viral RNA-binding protein that encapsulates the genome in a ratio of one N-protein per nine ribonucleotides. The long C-terminal IDPR and several shorter IDPRs make up roughly 31% of the protein (see Figure 4A). The N-protein, which is the most transcriptionally abundant protein during infection [123], has been shown to encapsulate the viral genomic RNA to protect it from nucleases and form a complex with the P-protein during replication [124]. The RABV P-protein binds to the RNA-free N°-protein through the N-terminus [100,125], which is predicted to be highly intrinsically disordered (see Figure 2A), whereas the N-terminal of the P-protein CTD is responsible for its interaction with the RNA-bound N-protein (see below).

The complex formed during replication, called a Negri body, is an inclusion in the host’s cytoplasm that is formed from interactions of the highly disordered C-terminal region of the N-protein with the intrinsically disordered regions of the P-protein [127]. The complex prevents non-specific RNA binding and the phosphorylation of the RNA [6]. The N-protein is predicted to have one MoRF (residues 406–411) located within the long C-terminal IDPR, which also includes a phosphoserine at position 389. A large region in the middle of the protein with little disorder suggests that the protein serves a largely structural purpose. The N-protein also functions to prevent the activation of the antiviral innate immune response receptor RIG-I-mediated antiviral response [19,20]. Although the actual information on the molecular mechanisms of this suppression is currently unavailable, it is tempting to hypothesize that the MoRF-containing IDPR found in the N-protein may serve to aid in suppressing the antiviral response. Figure 4A shows the results of the disorder prediction for the NCAP_RABVP protein. The crystal structure of the N-protein from the RABV strain ERA (which is 99.11% identical to the N-protein from the RABV strain PV) in complex with RNA was solved (PDB ID: 2GTT; [126]). Figure 4B shows that in this RABV nucleoprotein–RNA complex, the N-proteins are organized in an undecameric ring. In such a complex, the two core domains of the nucleoprotein clamp around the RNA at their interface and shield it from the environment [126]. Polymerization of the nucleoprotein is achieved by domain exchange between protomers, with flexible hinges allowing nucleocapsid formation. It is likely that the high structural flexibility and pliability of the C-terminal region of the protein, which is predicted to be highly intrinsically disordered, make it able to more tightly cover bound RNA. The nucleoprotein and the RNP core are able to adopt different conformations as a result at different periods of the viral cycle [127]. This important observation is illustrated in Figure 4C, which shows the structure of the N-protein protomer and demonstrates the presence of two “arms” in the structure (residues 6–28 and 349–414). Importantly, the C-terminal arm contains a MoRF. In addition to these functional arms, each protomer has several regions of missing electron density (residues 1–5, 104–117, 186–188, 374–397, and 449–450). Furthermore, according to the FuzDrop analysis, one can find four regions in the N-protein with context-dependent interactions (residues 105–115, 284–318, 367–396, and 398–411), which overlap, include, or are included in disordered/flexible regions of this protein (residues 103–111, 294–303, 361–400, and 403–428) (see Figure 4A).

Although, based on the FuzDrop analysis, the N-protein is expected to have a low probability of spontaneous LLPS (p_LLPS_ = 0.1405) and does not include any DPRs, this RNA-binding protein is invariantly present in Negri bodies (NBs) [100]. It is likely that the involvement of the N-protein in NB biogenesis is linked to the ability of this protein to bind both viral RNA and the P-protein. In fact, no NBs were found when limiting concentrations of one of these proteins were expressed in model experiments [100]. Furthermore, even when the RABV P- and N-proteins were expressed alone (i.e., without viral RNA), they were capable of forming NB-like structures [100]. These observations indicated that in the resulting N-P inclusions, the RABV N-protein is likely bound to cellular RNAs, forming N-RNA complexes similar to viral nucleocapsids [125]. It is known that different regions of the P-protein are utilized in the interaction with the N-protein, where the disordered N-terminal domain interacts with the RNA-free N°-protein, whereas the P-protein CTD binds to the RNA-associated N-protein [100,125]. As was already indicated, the positive patch within the N-terminal region of the P-protein CTD that is actually responsible for the interaction with the RNA-associated N-protein contains flexible/disordered residues, further emphasizing the potential role of structural disorder/flexibility in NB biogenesis.

### 3.4. Disorder of the G-Protein and Its Suggested Functional Consequences

The G-protein is a 524-residue-long type I transmembrane protein with a long extravirion region (residues 20–459), a transmembrane helix (residues 460–480), and an intravirion domain (residues 481–524). Being located on the surface of RABV particles, the G-protein controls receptor binding and the release of the viral ribonucleoprotein (RNP) in the cytoplasm via pH-dependent membrane fusion, thereby playing a crucial role in the cell entry and in vivo spread [128]. Furthermore, it was shown that the G-protein (in particular, its ectodomain) accumulates adaptive mutations that improve the release of infectious viral particles [129].

The G-protein is synthesized as a precursor with the N-terminal signal peptide (residues 1–19), which is removed during the maturation of this protein. Similar to the proteins found in the envelopes of other viruses, the RABV G-protein forms homotrimers on the surface of the virion that are responsible for the attachment of the virus to the host cellular receptors, such as the muscular form of the nicotinic acetylcholine receptor (nAChR), the neuronal cell adhesion molecule (NCAM), and the p75 neurotrophin receptor (p75NTR). There are approximately 400 such trimeric spikes, which are tightly arranged on the surface of the virus. The C-terminal domain of the G-protein (residues 258–505) is essential for trimer stability [8]. Figure 5A shows that the G-protein is mostly ordered and contains relatively few IDPRs, which is a characteristic feature of spike/glycoproteins of many other viruses. The most promising predicted disordered region is the C-terminal IDPR (residues 486–524; see Figure 5A), which corresponds to the intravirion domain of this protein (https://www.uniprot.org/uniprotkb/P08667/entry, accessed on 15 August 2022) that is engaged in the interaction with the matrix protein [13]. Several flexible regions located within the extravirion domain serve as aids in the interaction of the G-protein with surface molecules of the host cell [24]. There are three glycosylation cites in this protein (asparagine residues 56, 266, and 338) and a C-terminally located lipidation site, S-palmitoyl cysteine 479.

There is currently no structural information for the G-protein from the RABV strain PV. Therefore, we used SWISS-MODEL to create a homology model for this protein. Figure 5C shows a model for the 20–424 fragment of this protein that was generated using the known structure of the G-protein from the RABV strain CVS-11 (PDB ID: 6LGW [130]; UniProt ID: O92284) as a template, with sequence identity to the query G-protein of 91.48%. This structure is characterized by a highly elongated form and the presence of a C-terminal “arm” (residues 400–416). In the original structure of the G-protein from the RABV strain CVS-11, there are several regions of missing electron density (residues 21–24, 95–101, 133–143, 202–204, and 414–429).

Furthermore, the authors of this study present a structure for another form of this protein (PDB ID: 6LGX), which shows a different pattern of missing electron density regions (residues 21–25, 91–105, 131–147, 274–294, and 427–463). Since these two structures were resolved under different conditions (at ~pH-6.5 in the complex with a neutralizing antibody 523–11 (PDB ID: 6LGW) and at ~pH-8.0 in free form (PDB ID: 6LGX)), these observations suggest that this G-protein structure is characterized by its noticeable sensitivity to environmental conditions. The authors pointed out that the basic-to-acidic pH change results in large re-orientations of the three domains found in the G-protein from the RABV strain CVS-11, leading to concomitant domain-linker reconstructions that switch from a bent hairpin conformation into an extended conformation [130]. These low-pH-induced structural transitions within the domain-linker region are related to the functionality of the G-protein, playing important roles in G-protein-mediated membrane fusion [130]. Figure 5A,B show that the G-proteins from the RABV strains PV and CVS-11 are characterized by very similar disorder profiles, suggesting that the observations made for the structural peculiarities of the RABV G-protein from the CVS-11 strain are applicable to the G-protein from the RABV strain PV as well.

Similar to the N-protein, the glycoprotein of RABV is characterized by a low probability of spontaneous LLPS (p_LLPS_ = 0.1351) and does not include any DPRs but contains five regions with context-dependent interactions (residues 18–29, 218–223, 229–239, 434–441, and 502–508), which overlap, include, or are included in the disordered/flexible regions of this protein (residues 214–225, 422–442, and 486–524) (see Figure 5A).

### 3.5. Disorder of the L-Protein and Its Suggested Functional Consequences

With an amino acid sequence of 2124 residues, the L-protein is the longest protein in the RABV proteome. This protein is an RNA-directed RNA polymerase (RdRp) that catalyzes the transcription of viral mRNAs as well as their polyadenylation and capping. It has several functional regions, such as an RdRp catalytic domain (residues 611–799), a mononegavirus-type SAM-dependent 2′-O-MTase domain (residues 1674–1871) included in the C-terminal region that is involved in the interaction with the P-protein (residues 1562–2127) and contains a disorder-based interaction site, and a MoRF (residues 1631–1638). In addition to interactions with the P-protein, the L-protein may form homodimers. Although the L-protein includes many disordered or flexible regions, its overall intrinsic disorder level is relatively low (see Figure 6A).

It is possible that the need for interplay between ordered and disordered features in this protein reflects its purpose to serve as a low-fidelity viral polymerase, as intrinsic disorder and structural flexibility likely define the lower fidelity of the polymerase, which results in a high mutation rate and therefore higher flexibility in the ability of the virus to adapt to host defenses [7]. The polymerase activity of the L-protein depends on the identity of residues upstream of the protein, as well as the identity of C-terminal residues [131]. Figure 6C shows the structural model for the L-protein from the RABV strain PV built using the cryo-EM structure of the large structural protein from the RABV strain SAD B19 (sequence identity: 98.68%) complexed with a fragment of the P-protein (PDB ID: 6UEB, [132]; UniProt ID: P16289) as a template. Due to their high sequence similarity, the disorder profiles of the L-proteins from RABV strains SAD B19 and PV are almost identical (cf. Figure 6A,B). Since in the aforementioned cryo-EM structure of the L-P complex, residues 1–27 of the L-protein from the RABV strain SAD B19 constitute a region of missing electron density, it is likely that this N-terminal region is disordered in the L-protein from the RABV strain PV as well.

**Figure 6 biomolecules-12-01436-f006:**
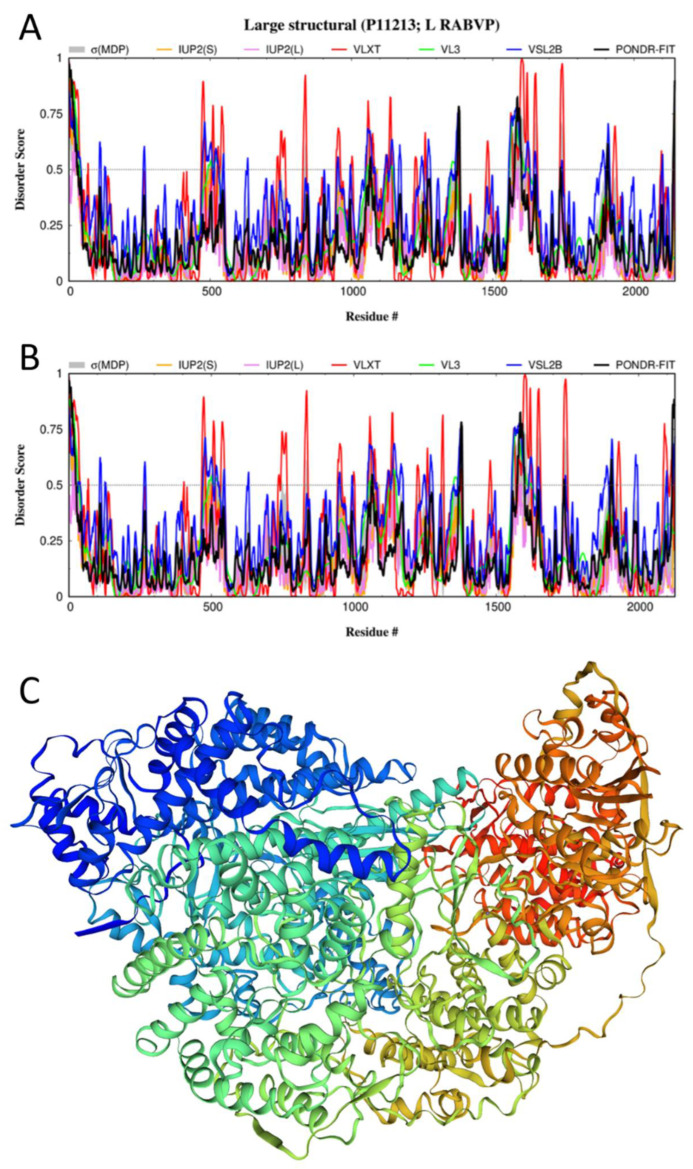
Structure and disorder in the L-protein from RABV. (**A**) Intrinsic disorder profile generated for the L-protein from the RABV strain VP (UniProt ID: P11213) by RIDAO. (**B**) Intrinsic disorder profile generated for the L-protein from the RABV strain SAD B19 by RIDAO (UniProt ID: P16289). (**C**) A structural model for the L-protein from the RABV strain PV built by SWISS-MODEL [97] using the cryo-EM structure of the large structural protein from the RABV strain SAD B19 (sequence identity: 98.68%) complexed with the fragment of the P-protein (PDB ID: 6UEB, [132]; UniProt ID: P16289) as a template.

FuzDrop indicated that the L-protein has a low p_LLPS_ value of 0.11860 and does not contain any DPRs. However, this protein possesses 24 regions with context-dependent interactions (residues 5–12, 104–109, 416–434, 456–480, 507–514, 566–571, 672–677, 821–833, 852–859, 878–899, 903–911, 924–929, 982–987, 1014–1022, 1152–1157, 1232–1238, 1383–1388, 1390–1399, 1562–1570, 1655–1667, 1731–1753, 1980–1985, 2086–2092, and 2127–2132). Many of these regions are related to the IDPRs identified in the L-protein (e.g., residues 1–42, 108–114, 466–480, 504–512, 827–839, 1555–1599, 1642–1656, 1735–1750, 2093–2096, and 2137–2142).

### 3.6. Intrinsic Disorder in Human Proteins Interacting with the RABV Proteins

#### 3.6.1. Host Interactors of the P-Protein

In addition to being involved in interactions with the RABV L- and N-proteins and viral ribonucleocapsids, the P-protein is known to bind to a multitude of host proteins, such as dynein light-chain 1 and 2 (DYNLL1 (UniProt ID: P63167) and DYNLL2 (UniProt ID: Q96FJ2), respectively) [133], as well as host signal transducer and activator of transcription 1-alpha/beta and signal transducer and activator of transcription 2 (STAT1 (UniProt ID: P42224) and STAT2 proteins (UniProt ID: P52630), respectively) [16,17,101,102,103,104], promyelocytic leukemia (PML) protein (UniProt ID: P29590) [110], the ribosomal protein L9 (UniProt ID: Q02878) [15], STAT3 (UniProt ID: P40763), nucleolin (NCL; UniProt ID: P19338), focal adhesion kinase (FAK; UniProt ID: Q05397), Janus kinase 1 (JAK1; UniProt ID: P23458), inhibitor of nuclear factor kappa-B kinase subunit epsilon (IKKε; UniProt ID: Q14164), Beclin-1 (BECN1; UniProt ID: Q14457), tubulin alpha (TUB-α; UniProt ID: Q71U36 for tubulin alpha-1A chain), tubulin beta (TUB-β; UniProt ID: Q9H4B7 for tubulin beta-1 chain), ATP-binding cassette sub-family E member 1 (ABCE1; UniProt ID: P61221), T-complex protein 1 subunit gamma (CCTγ; UniProt ID: P49368), Hsp90 co-chaperone Cdc37 (CDC37; UniProt ID: Q16543), and heat shock protein HSP 90-alpha (Hsp90AA1; UniProt ID: P07900) [71]. Furthermore, the P-protein can interact with complex I in mitochondria, leading to mitochondrial dysfunction, the increased generation of reactive oxygen species (ROS), and oxidative stress [134]. Being the largest and most complicated component of the respiratory chain, complex I contains 45 subunits [135]. Unfortunately, the exact targets of the P-protein within this complex are unknown. Therefore, the proteins forming complex I were not included in the analysis.

#### 3.6.2. Host Interactors of the M-Protein

Several human proteins that interact with the RABV M-protein were established. The list includes RelAp43 (which is a splicing variant of RelA (UniProt ID: Q04206) and other proteins in the NF-*κ*B pathway [23], as well as V-type proton ATPase catalytic subunit A (ATP6V1A; UniProt ID: P38606), E3 ubiquitin-protein ligase NEDD4 (NEDD4; UniProt ID: P46934), transcriptional coactivator YAP1 (UniProt ID; P46937), eukaryotic translation initiation factor 3 subunit H (EIF3H, UniProt ID: O15372), JAK1 (UniProt ID: P23458), and STAT1 (UniProt ID: P63167) [71]. Note that the M-protein shares two human interactors (JAK1 and STAT1) with the P-protein [71].

#### 3.6.3. Host Interactors of the N-Protein

Very few host proteins were shown to act as physical partners of the RABV N-protein. These include heat shock 70 kDa proteins 1A (HSPA1A, UniProt ID: P0DMV8) and 1B (HSPA1B, UniProt ID: P0DMV9), prefoldin subunit 1 (PFDN1, UniProt ID: O60925), and CCTγ (UniProt ID: P49368), with CCTγ being a shared partner for the RABV P- and N-proteins [71].

#### 3.6.4. Host Interactors of the G-Protein

The major biological function of the G-protein is cell entry via interaction with RABV entry receptors, such as nicotinic acetylcholine receptors (nAChR), the neuronal cell adhesion molecule (NCAM1; UniProt ID: P13591), the low-affinity p75 neurotrophin receptor (p75NTR, also known as tumor necrosis factor receptor superfamily member 16; UniProt ID: P08138), and metabotropic glutamate receptor subtype 2 (mGluR2; UniProt ID: Q14416) [136,137,138,139]. nAChRs are assemblies of five subunits, which are arranged symmetrically around a central pore. It is known that the neuronal subtypes of nAChRs (which serve as the receptors of the RABV G-protein) exist as various homomeric or heteromeric (at least one α and one β) combinations of twelve different nicotinic receptor subunits: α_2_−α_10_ and β_2_−β_4_ (with some of the neuronal nAChR subtypes being (α_4_)_3_(β_2_)_2_, (α_4_)_2_(β_2_)_3_, (α_3_)_2_(β_4_)_3_, α_4_α_6_β_3_(β_2_)_2_, and (α_7_)_5_). Therefore, there are multiple possibilities for the binding of the G-protein to nAChR. Physical interactions of nicotinic acetylcholine receptor alpha 1 (nAChR α1 or CHRNA1; UniProt ID: P02708) [140] and nAChr α7 (CHRNA7, UniProt ID: P36544) [141] with the RABV G-protein were demonstrated. Furthermore, the G-protein can bind host microtubule-associated serine/threonine-protein kinases 1 and 2 (MAST1 (UniProt ID: Q9Y2H9) and MAST2 (UniProt ID: Q6P0Q8), respectively), tyrosine-protein phosphatase non-receptor type 4 (PTPN4; UniProt ID: P29074), disks large homolog 2 (DLG2; UniProt ID: Q15700), multiple PDZ domain protein (MPDZ; UniProt ID: O75970), and synaptosomal-associated protein 25 (SNAP25; UniProt ID: P60880) [71].

#### 3.6.5. Host Interactors of the L-Protein

The only human partner of the L-protein is DYNLL1, which is shared with the P-protein [71].

#### 3.6.6. Prevalence of Intrinsic Disorder in Human Proteins Interacting with RABV

Detailed characterizations of the prevalence of functional disorder in each of the human proteins interacting with the RABV P-, M-, N-, G-, and L-proteins are systemized in Appendix A, respectively. The corresponding analyses were conducted using a set of computational tools, such as RIDAO, STRING, D^2^P^2^, and AlphaFold. This revealed a very high level of disorder in the majority of the proteins from this dataset, with the entire set being characterized by a mean PPIDR of 41.6 ± 20.9% (as evaluated using the outputs of the PONDR^®^ VSL2 predictor, which is one of the most accurate stand-alone disorder predictors [142,143]).

Appendix A show that all of these proteins contain multiple IDPRs of various lengths. Many proteins contain multiple MoRFs, and almost all human proteins in this dataset are densely decorated by a multitude of different PTMs. These observations suggest that intrinsic disorder in this these proteins might be related to their functionality, playing a role in their binding promiscuity, as evidenced by dense PPI networks centered on these proteins.

These observations are further illustrated in Figure 7, which presents the global intrinsic disorder characteristics of 37 human proteins. In fact, Figure 7A shows there is no single protein in this dataset that could be classified as mostly ordered, whereas 25 proteins (67.6%) are expected to be mostly disordered (i.e., their PPIDR exceeds 30%). This classification is accepted in field practice to group proteins based on their PPIDR values [87], where proteins with PPIDR < 10% are considered ordered or mostly ordered; proteins with 10% ≤ PPIDR < 30% are considered moderately disordered; and proteins with PPIDR ≥ 30% are considered highly disordered [87]. Furthermore, PPIDR values for 13 proteins (NCL (P19338; 86.20%), YAP1 (P46937; 82.94%), SNAP25 (P60880; 78.16%), MAST2 (Q6P0Q8; 73.75%), MAST1 (Q9Y2H9; 68.60%), CDC37 (Q16543; 65.08%), RelAp43 (Q04206; 64.61%), NEDD4 (P46934; 62.17%), L9 (Q02878; 61.46%), BECN1 (Q14457; 58.00%), p75NTR (P08138; 57.14%), PML (P29590; 54.06%), and PFDN1 (O60925; 50%)) exceed 50%, indicating that 35.1% of human proteins interacting with RABV are expected to be extremely disordered. These levels of disorder in human RABV interactors are comparable to those observed in the entire human proteome, where out of 20,317 proteins, 12,363 proteins (60.8%) and 7590 (37.3%) are characterized by PPIDR ≥ 30% and PPIDR ≥ 50%, respectively.

Figure 7B shows the ΔCH-ΔCDF plot (a combined binary predictor of intrinsic disorder) that verifies a global prevalence of intrinsic disorder in 37 human proteins interacting with the RABV proteins. The ΔCH-ΔCDF plot provides the means for the evaluation of the flavors of intrinsic disorder. Figure 7B shows that quadrant Q1 (bottom right corner) contains 22 proteins that are predicted to be ordered by both predictors; quadrant Q2 (bottom left corner) includes 10 proteins, which are predicted to be ordered/compact by the CH-plot and disordered by CDF (i.e., it contains either molten globular proteins, which are compact, but without unique 3D structures, or hybrid proteins containing comparable levels of ordered and disordered residues); and quadrant Q3 (top left corner) includes 4 highly disordered regions (native coils or native pre-molten globules), which are predicted to be disordered by both predictors. Finally, one protein in quadrant Q4 (top right corner) is predicted to be disordered by the CH-plot and ordered by the CDF-plot.

Therefore, 15 human proteins that interact with the RABV proteins are predicted to contain very noticeable levels of disorder (i.e., they are located outside quadrant Q1). This correlates well with the results shown in Figure 7A, where 13 proteins are located within the dark-red segment.

The apparent discrepancies between the data shown in Figure 7A,B are rooted in the principle differences in the tools utilized for these analyses, where Figure 7A represents the outputs of the per-residue predictor, whereas Figure 7B reports data generated by the so-called binary predictors; i.e., tools that classify query proteins as mostly ordered or mostly disordered. Obviously, mostly ordered proteins might contain noticeable levels of disordered residues, whereas mostly disordered proteins might possess noticeable levels of ordered residues.

Not surprisingly, all 37 proteins in the analyzed set were found to form a rather dense PPI network (see Figure 8A), in which, on average, each protein interacts with at least 11 partners.

In this intraset PPI network, the five most significantly enriched biological processes were viral process (GO:0016032; *p* = 3.49 × 10^−8^), immune effector process (GO:0002252; *p* = 0.00030), interspecies interaction between organisms (GO:0044419; *p* = 0.00030), intracellular transport (GO:0046907; *p* = 0.00057), and cellular component organization (GO:0016043; *p* = 0.00057). Among the molecular functions, the five most significantly enriched were ubiquitin-like protein ligase binding (GO:0044389; *p* = 7.02 × 10^−7^), protein binding (GO:0005515; *p* = 1.22 × 10^−6^), ubiquitin protein ligase binding (GO:0031625; *p* = 6.83 × 10^−5^), enzyme binding (GO:0019899; *p* = 8.27 × 10^−5^), and binding (GO:0005488; *p* = 8.27 × 10^−5^). Finally, the five most significantly enriched cellular components were postsynapse (GO:0098794; *p* = 1.79 × 10^−6^), cytosol (GO:0005829; *p* = 1.79 × 10^−6^), synapse (GO:0045202; *p* = 2.96 × 10^−5^), cell junction (GO:0030054; *p* = 0.00010), and neuron projection (GO:0043005; *p* = 0.00017).

We also analyzed the set-based interactivity of these human proteins. To this end, STRING was used to generate a PPI network that includes first-shell interactors (see Figure 8B). The resulting highly interlinked interactome includes 536 proteins connected by 11,358 interactions. Therefore, this interactome is characterized by an average node degree of 42.4, and it shows an average local clustering coefficient of 0.631. The expected number of interactions for the set of proteins of its size is 5.089, indicating that this PPI network, centered on human proteins interacting with the RABV proteins, has significantly more interactions than expected (PPI enrichment *p*-value is < 10^−16^). Looking at the functional enrichment of this network based on Gene Ontology (GO) terms revealed that the five most significantly enriched biological processes were interspecies interaction between organisms (GO:0044419; *p* = 6.95 × 10^−102^), viral process (GO:0016032; *p* = 1.60 × 10^−99^), symbiotic process (GO:0044403; *p* = 1.60 × 10^−99^), translational initiation (GO:0006413; *p* = 7.45 × 10^−88^), and cellular response to organic substance (GO:0071310; *p* = 6.37 × 10^−76^). The five most significantly enriched functions were structural constituent of ribosome (GO:0003735; *p* = 5.85 × 10^−66^), protein binding (GO:0005515; *p* = 3.38 × 10^−57^), binding (GO:0005488; *p* = 2.02 × 10^−52^), enzyme binding (GO:0019899; *p* = 3.08 × 10^−46^), and RNA binding (GO:0003723; *p* = 2.09 × 10-^37^). Among the cellular components, the five most significantly enriched were cytosol (GO:0005829; *p* = 6.45 × 10^−106^), protein-containing complex (GO:0032991; *p* = 2.61 × 10^−82^), cytosolic ribosome (GO:0022626; *p* = 5.33 × 10^−81^), cytoplasm (GO:0005737; *p* = 6.02 × 10^−68^), and ribosomal subunit (GO:0044391; *p* = 5.65 × 10^−65^).

Although a detailed description of the potential roles of intrinsic disorder in the functionality of human proteins shown to interact with the RABV proteins is outside the scope of this study, it is clear that all of the proteins analyzed here (i.e., RABV N-, L-, P-, M-, and G-proteins and 37 human proteins) contain very significant levels of intrinsic disorder. This is further illustrated in Figure 9, showing experimentally identified and validated interactions between the five RABV proteins, G (glycoprotein), N (nucleoprotein), L (RNA-dependent polymerase), P (phosphoprotein), and M (matrix protein), and 37 human proteins. This diagram clearly shows that most of the proteins (viral and human) in this network are “red” (highly disordered), and there are no “blue” (mostly ordered) proteins, suggesting the importance of intrinsic disorder for RABV infection.

## 4. Conclusions

All Rabies lyssavirus PV proteins contain IDPRs, most of which are expected to aid in the flexibility of the virus and its ability to evade host antiviral defenses. All human proteins found to be RABV interactors also contain high levels of intrinsic disorder, with most of these proteins being highly disordered. This disorder-centric layer of complexity in RABV and its interaction with host proteins adds a new angle to the search for potential targets for anti-rabies drugs. Once the virus has infected a host cell, there are virtually no effective treatments available that can be used to destroy the virus. Although the modern RABV vaccine has largely eradicated the virus in developed countries, many people around the world are unable to seek treatment until the virus has already crossed the blood–brain barrier. Without the immediate usage of the vaccine upon infection, the RABV-related mortality rate is close to 100%. The advent of bioinformatics approaches in a clinical setting has inspired many developments that have led to the creation of new drugs. It is likely that targeting regions of intrinsic disorder within the viral proteins or host proteins that directly interact with the RABV proteins will help in creating novel drugs that can target the virus once it has infected the central nervous system.

## Figures and Tables

**Figure 1 biomolecules-12-01436-f001:**
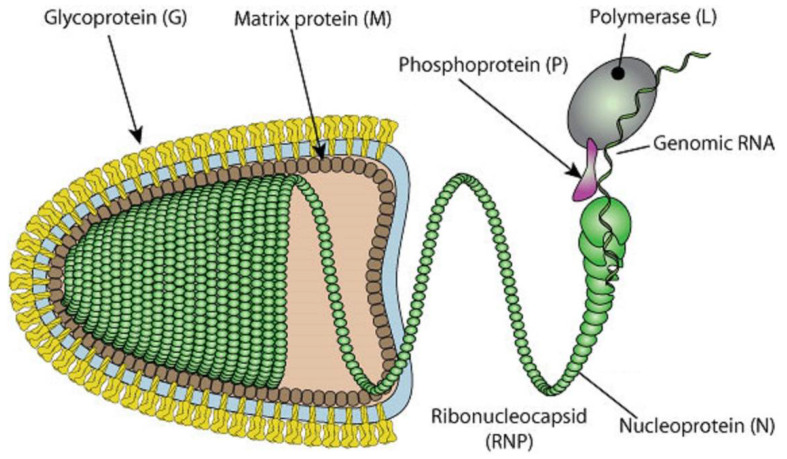
Structural features of RABV. In mature RABV, the nucleoprotein, phosphoprotein, and viral polymerase envelop the genomic RNA in a structure known as the ribonucleocapsid (RNP). The matrix protein surrounds the RNP and determines the shape of the virus. The matrix protein also anchors the glycoprotein to the envelope [10] (original source of the image: Philippe Le Mercier, SIB Swiss Institute of Bioinformatics).

**Figure 2 biomolecules-12-01436-f002:**
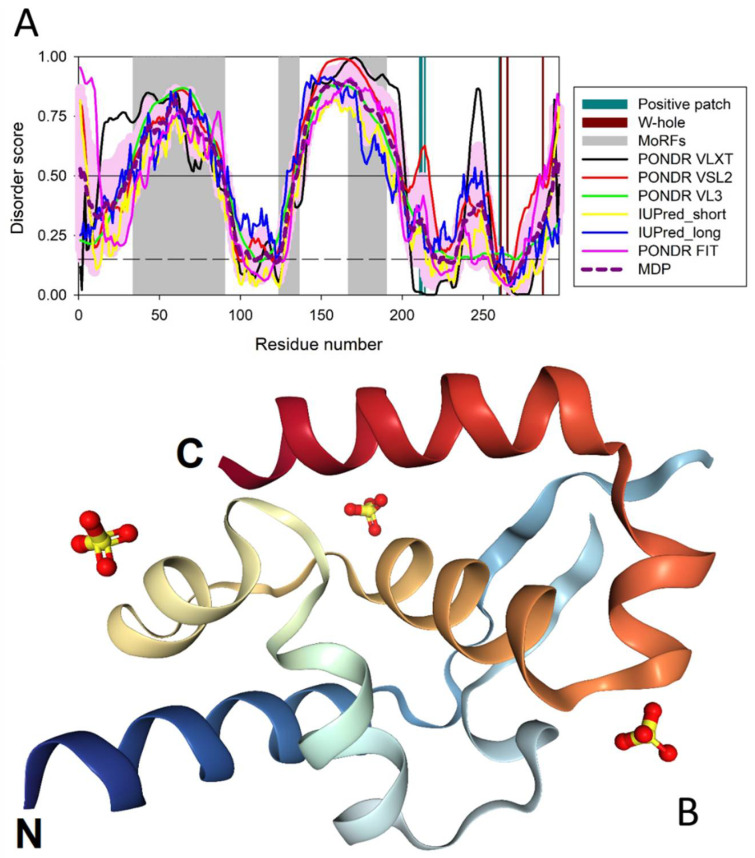
Structure and disorder in the RABV P-protein (UniProt ID: P06747). (**A**) Intrinsic disorder profile generated using data aggregated by the RIDAO platform. The profile also contains disorder/flexibility-related functional annotations and shows three predicted MoRFs (gray-shaded areas), positive patch residues (dark-cyan vertical bars), and W-hole residues (dark-red vertical bars). (**B**) Crystal structure of the C-terminal region of the RABV P-protein (residues 192–295) (PDB ID: 3OA1).

**Figure 4 biomolecules-12-01436-f004:**
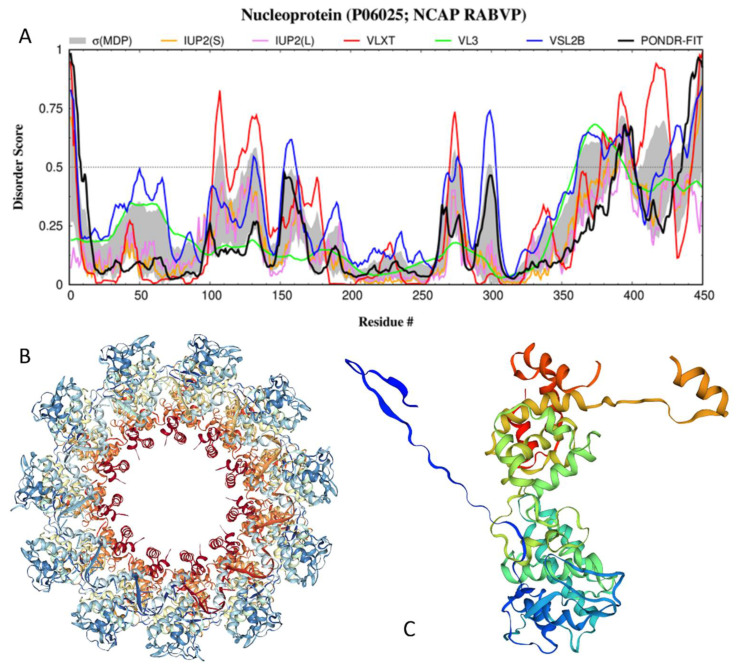
Structure and disorder in the N-protein from the RABV strain PV (UniProt ID: P06025). (**A**) Intrinsic disorder profile generated for the M-protein from the RABV strain VP by RIDAO. (**B**) Crystal structure of the N-protein from the RABV strain ERA (which is 99.11% identical to the N-protein from the RABV strain PV) in complex with RNA (PDB ID: 2GTT; [126]), where the protomers of the N-protein are organized in an undecameric ring. (**C**) Crystal structure of the N-protein protomer computationally taken out of the undecameric homo-oligomer and demonstrating the presence of two “arms” in the structure (residues 6–28 and 349–414).

**Figure 5 biomolecules-12-01436-f005:**
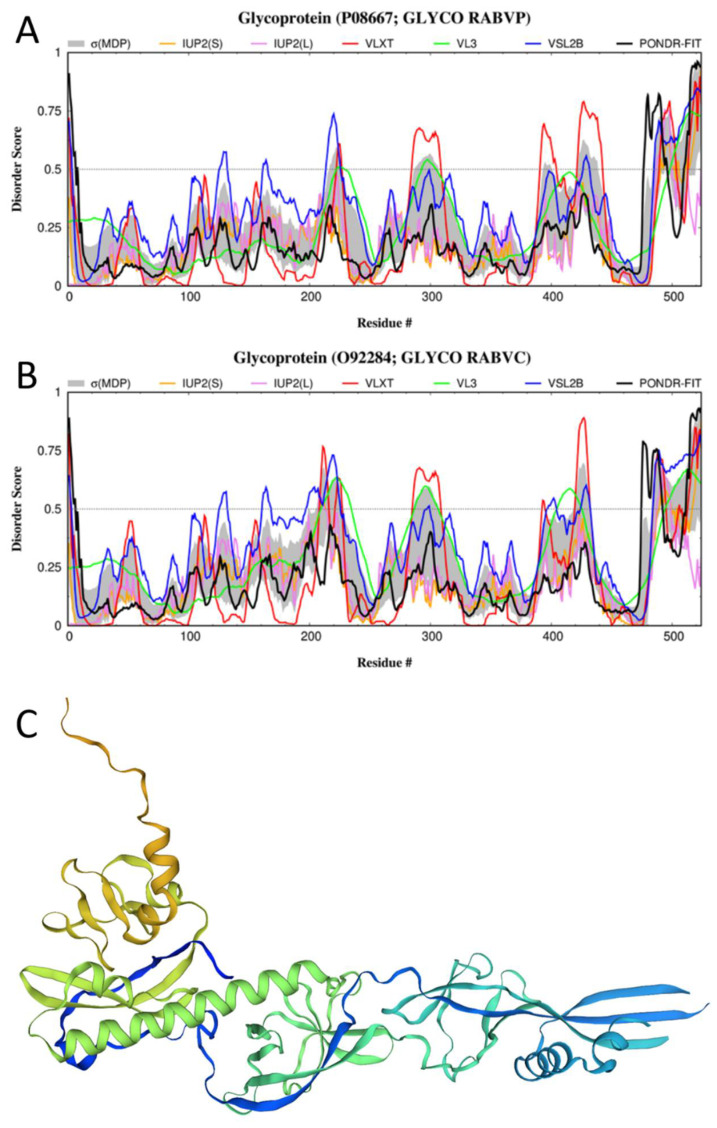
Structure and disorder in the G-protein from RABV. (**A**) Intrinsic disorder profile generated for the G-protein from the RABV strain VP (UniProt ID: P08667) by RIDAO. (**B**) Intrinsic disorder profile generated for the G-protein from the RABV strain CVS-11 by RIDAO (UniProt ID: O92284). (**C**) A structural model for the 20–424 fragment of the G-protein from the RABV strain VP generated by SWISS-MODEL [97] using the known structure of the G-protein from the RABV strain CVS-11 (PDB ID: 6LGW [130]; UniProt ID: O92284) as a template, with sequence identity to the query G-protein of 91.48%.

**Figure 7 biomolecules-12-01436-f007:**
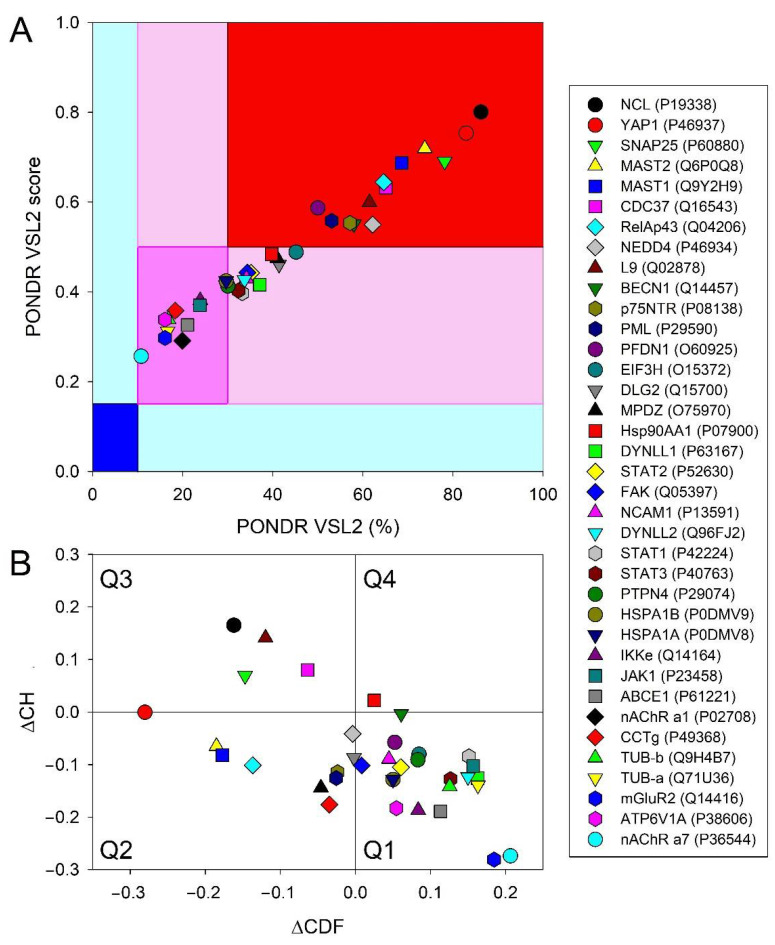
Evaluation of global disorder in 37 human proteins interacting with the RABV proteins. (**A**) PONDR^®^ VSL2 output for 37 human proteins. The PONDR^®^ VSL2 score is the average disorder score (ADS) for a protein. PONDR^®^ VSL2 (%) is the percent of predicted disordered residues (PPDR). i.e., residues with disorder scores above 0.5. Color blocks indicate regions in which proteins are mostly ordered (blue and light blue), moderately disordered (pink and light pink), or mostly disordered (red). If the two parameters agree, the corresponding part of the background is dark (blue or pink), whereas light blue and light pink reflect areas in which only one of these criteria applies. (**B**) Charge-hydropathy and cumulative distribution function (CH-CDF) plot. The Y-coordinate is calculated as the distance of the corresponding protein from the boundary in the CH plot. The X-coordinate is calculated as the average distance of the corresponding protein’s CDF curve from the CDF boundary. The quadrant in which the protein is located determines its classification. Q1, protein predicted to be ordered by the CH-plot and CDF. Q2, protein predicted to be ordered by the CH-plot and disordered by the CDF-plot. Q3, protein predicted to be disordered by the CH-plot and CDF. Q4, protein predicted to be disordered by the CH-plot and ordered by CDF.

**Figure 8 biomolecules-12-01436-f008:**
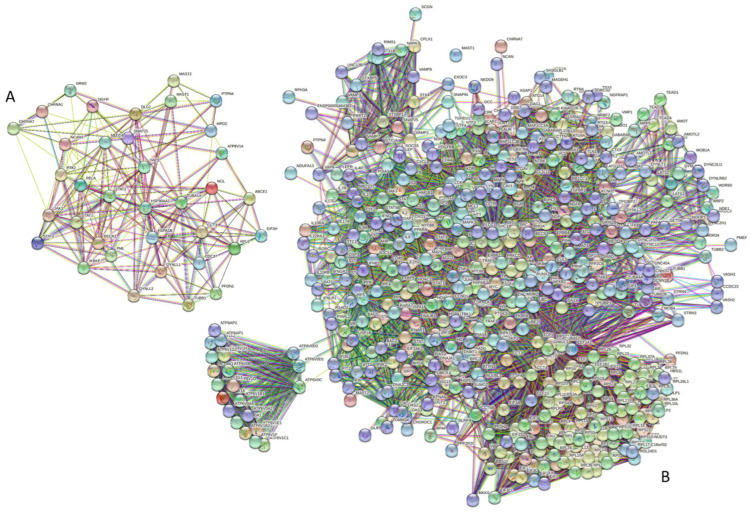
Intraset and set-based interactivity of human proteins engaged in interactions with the RABV proteins. (**A**) STRING-generated PPI network with the analyzed set of human proteins. To include all proteins in the network, a low confidence of 0.15 was used as the minimum required interaction score in this case. This network includes 36 proteins linked by 204 interactions. The resulting average node degree of this network is 11.3, and its average local clustering coefficient (which defines how close its neighbors are to being a complete clique; the local clustering coefficient is equal to 1 if every neighbor connected to a given node Ni is also connected to every other node within the neighborhood, and it is equal to 0 if no node that is connected to a given node Ni connects to any other node that is connected to Ni) is 0.578. Since the expected number of edges in a network of the same size for proteins randomly selected from the human proteome is 128, this network is characterized by a PPI enrichment *p*-value of 4.7 × 10^−10^. (**B**) The STRING-generated PPI network centered on human proteins interacting with the RABV proteins. Note that the number of interactors in STRING is limited to 500. This network, generated with a high confidence score of 0.7, includes 536 proteins connected by 11,358 interactions. The average node degree and average local clustering coefficient of this PPI are 42.4 and 0.631, respectively; its PPI enrichment *p*-value is <10^−16^.

**Figure 9 biomolecules-12-01436-f009:**
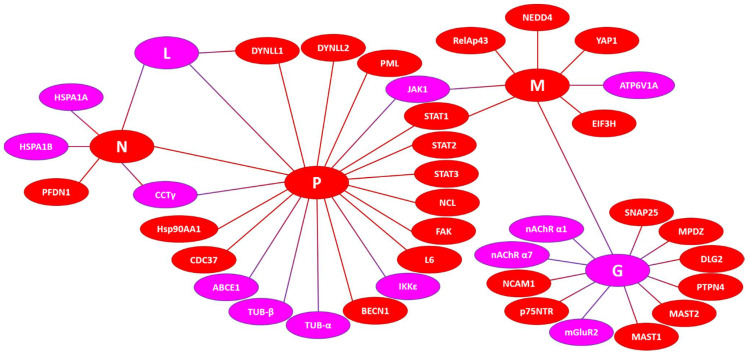
Disordered interactome of the RABV N-, L-, P-, M-, and G-proteins. Proteins are colored based on their PPIDR values, with highly and moderately disordered proteins being shown by red and pink colors, respectively. Note that none of the proteins in this diagram is classified as mostly ordered (there are no proteins colored in blue).

**Table 1 biomolecules-12-01436-t001:** Proteins in the rabies PV proteome and their evaluated percentages of intrinsically disordered residues as determined by the averaged predicted disorder.

Protein	UniProt Entry ID [70]	Protein Length (Residues)	Longest Disordered Region (Residues)	Percent of Disordered Residues
P (Phosphoprotein)	P06747	297	87	67.3%
M (Matrix protein)	P08671	202	54	43%
N (Nucleoprotein)	P06025	450	93	30.6%
G (Glycoprotein)	P08667	524	49	27%
L (Large protein)	P11213	2142	105	23%

## Data Availability

Data are contained within the article or Appendix A.

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
