# Peer review of "Looking at the Pathogenesis of the Rabies Lyssavirus Strain Pasteur Vaccins through a Prism of the Disorder-Based Bioinformatics"

_biomolecules, 2022, doi:10.3390/biom12101436_

Round 1

Reviewer 1 Report

An interesting and well-written research article entitled "Looking at the Pathogenesis of the Rabies Lyssavirus Strain Pasteur" by Surya Dhulipala and Vladimir N. Uversky is dedicated to the computational analysis of the RABV proteins and human proteins interacting with RABV proteins. The authors used an intrinsic disorder perspective to look at the multifunctionality of RABV proteins and to deliver a disorder-centric view of human proteins that were experimentally shown to interact with RABV. Therefore, importance of this study is in its unique view on the major viral and host players associated with the RABV infection. It is well known that although the timely conducted postexposure prophylaxis after potential exposures but before symptoms start (that includes a dose of human rabies immune globulin (HRIG) and rabies vaccine given on the day of the rabies exposure, and then a dose of vaccine given again on days 3, 7, and 14) provides efficient means to prevent rabies, a person who did not receive the appropriate medical care on a timely manner, will develop disease, which is virtually always resulting in death. In fact, rabies is virtually 100% fatal after onset of symptoms. Therefore, reported in this study analysis is very important, as it potentially opens some new perspectives in the development of novel therapeutic approaches to RABV infection and might help with finding a novel cure for this fatal but preventable viral disease, rabies. 

Author Response

Reviewer #1

An interesting and well-written research article entitled "Looking at the Pathogenesis of the Rabies Lyssavirus Strain Pasteur" by Surya Dhulipala and Vladimir N. Uversky is dedicated to the computational analysis of the RABV proteins and human proteins interacting with RABV proteins. The authors used an intrinsic disorder perspective to look at the multifunctionality of RABV proteins and to deliver a disorder-centric view of human proteins that were experimentally shown to interact with RABV. Therefore, importance of this study is in its unique view on the major viral and host players associated with the RABV infection. It is well known that although the timely conducted postexposure prophylaxis after potential exposures but before symptoms start (that includes a dose of human rabies immune globulin (HRIG) and rabies vaccine given on the day of the rabies exposure, and then a dose of vaccine given again on days 3, 7, and 14) provides efficient means to prevent rabies, a person who did not receive the appropriate medical care on a timely manner, will develop disease, which is virtually always resulting in death. In fact, rabies is virtually 100% fatal after onset of symptoms. Therefore, reported in this study analysis is very important, as it potentially opens some new perspectives in the development of novel therapeutic approaches to RABV infection and might help with finding a novel cure for this fatal but preventable viral disease, rabies. 

RESPONSE: We are thankful to this reviewer for warm words and high evaluation of our manuscript.

Author Response

Reviewer #2

The article by Dhulipala and Uversky examines the predicted disorder in the proteome of RABV and its human proteome interaction network. The conclusions claimed in this article are not supported, as a logical link between a predicted disorder profile and function is never made. Additionally, in many places the citations used to suggest a particular function are inappropriate as the cited article does not refer to said function. Please see below for additional details.

RESPONSE: We are thankful to this reviewer for critical reading of our manuscript and for providing constructive critiques. We tried to address all the queries and revised manuscript accordingly.

Major comments:

The abstract states, “The study suggests that the high levels of intrinsic disorder in the phosphoprotein (P-protein) and nucleoprotein (N-protein) allow them to participate in creation of the Negri bodies and help this virus suppress the antiviral immune response in the host cells. Additionally, the study suggests that there is a link between disorder in the matrix (M) protein and the modulation of viral transcription. The disordered regions in the M protein have a possible role in initiating viral budding within the cell.” However, just showing that a protein has x% disorder does not imply that the disorder regions are important for each of the different functions laid out. Thus, this is misleading because the study does not provide a direct link to disorder playing a specific role in any of these functions. For a link between disorder and function to be made more than just a predicted disorder profile has to be shown. Thus, such statements throughout the article need to be greatly toned down in terms of this study providing evidence for functional relevance.

RESPONSE: Thank you for pointing this out. Although we are respectively disagree with the claim that we overstated and over interpreted our observations, as we are indicating that our “study suggests” rather than provide the unambiguous proof or unquestionable evidence. Nowhere in the manuscript have we claimed that our study is “providing evidence for functional relevance”. However, to address this concern, we further modified text of abstract to read: “This study aims to make use of disorder-based proteomics and bioinformatics to determine the potential impact that intrinsically disordered protein regions (IDPRs) in the proteome of the rabies virus might have on the infectivity and lethality of the disease. This study uses the proteome of Rabies Lyssavirus (RABV) strain Pasteur Vaccins (PV), one of the best understood strains due to its use in the first rabies vaccine, as a model. Data reported in this study are in line with the hypothesis that the high levels of intrinsic disorder in the phosphoprotein (P-protein) and nucleoprotein (N-protein) allow them to participate in creation of the Negri bodies and might help this virus to suppress the antiviral immune response in the host cells. Additionally, the study suggests that there could be a link between disorder in the matrix (M) protein and the modulation of viral transcription. The disordered regions in the M protein might have a possible role in initiating viral budding within the cell.”

On pg. 7 it states, “Additionally, this high disorder content suggests that the mechanism of action for the P-protein to suppress the Type-I interferon-mediated immune response may be to use its disordered status for interaction with the STAT proteins of the cells…The IDPRs that were found in this protein corresponded to ligand binding sites, which supports the idea that the IDPRs interact with the STAT1 and STAT2 proteins”. How does having high disorder imply that disorder is important for a specific interaction with another protein, particularly the STAT proteins? How does the IDP having a general “ligand binding site” imply that the IDP interacts with STAT1 and STAT2? These statements need to be explained and unless there is a specific feature the authors can point out this seems like too big of a jump. Furthermore, the citation (Wiltzer et al.) used to support an interaction between the P-protein and STAT proteins shows that the interaction is driven by W-hole residues in the folded region of the P-protein and ∆C30, which removes this folded domain, eliminates interactions between the P-protein and the STAT proteins. Thus, evidence supporting that the IDRs are important for interactions with STAT proteins is severely lacking.

RESPONSE: In this section of the manuscript we discuss not only regions that have high levels of disorder, but also talk about MoRFs, which are disorder-based binding sites that are expected to undergo the disorder-to-order transition at interaction with binding partners. This information is clearly stated in the original version of the manuscript, but is ignored by the reviewer. However, we appreciate the important notion of the reviewer that STAT proteins interact with the ordered domain of the RABV P-protein. To address this point, we conducted more detailed analysis of the peculiarities of disorder distribution within this protein. We found that the residues of the W-hole (C261, W265, and M287) are predicted as flexible, with highest flexibility level was ascribed to the M287 residue. We added the corresponding clarification to the revised manuscript:

Additionally, the high disorder and structural flexibility content of the RABV P-protein suggests that the mechanism of action of this protein to suppress the Type-I interferon-mediated immune response may be based on the utilization of its disordered (or flexible) regions for interaction with the STAT proteins of the cells. In fact, although the RABV P-protein binding site responsible for the STAT1/2 interaction is located within the ordered C-terminal domain (CTD, residues 186-297 [16,17,100-103]), the residues that made most contribution to this interaction and that form the so-called W-hole (C261, W265, and M287) were predicted as flexible by at least one of the disorder predictors used in this study, with the highest level of structural flexibility being expected for the residue M287, which is 100% conserved among most of the lyssavirus P-proteins [16], and which shows the mean disorder score of 0.31±0.12 (ranging from 0.16 to 0.5, as per the outputs of IUPred_short and PONDR® VSL2, respectively (see Figure 2A). Similarly, the positive patch (residues K211, K212, K214, and R260), which are 100% conserved in the lyssavirus P-proteins and known to be responsible for interaction with the N-protein [16] are predicted as flexible/disordered as well (see Figure 2A). This, again, suggests that structural flexibility plays a role in interaction of the ordered CTD with partner proteins, including STAT1 and STAT2. It was pointed out that because of these C-terminal domain-driven interactions of the RABV P-protein (CTD, residues 186-297 [100-103]) with host STAT proteins, the P-protein represents the major interferon antagonist of the lyssavirus, thereby affecting the type-I interferon (IFNα/β)-mediated innate immune response [16]. It was also pointed out that interaction of the RABV P-protein with STATs is crucial for the development of the lethal rabies disease [16].

Please note that Figure 2 was also modified to reflect these observations.

On pg. 9 FuzDrop scores are given for P-protein isoforms, yet the reader is given no context about what these scores mean. Does 0.52 mean it has a 50% chance of undergoing LLPS? For these scores to have meaning for the reader some scale information is needed such as what is the average score for proteins that are known to undergo LLPS and where do the P-protein scores fall on the distribution of scores of a large set of proteins? Do these values actually support the claim that the protein “has a strong LLPS potential”?

RESPONSE: Thank you for pointing this out. The corresponding clarifications are added to the revised manuscript. The text now includes the following addition:

According to the FuzDrop developers, proteins with pLLPS ≥ 0.60 are droplet-drivers, which can spontaneously undergo LLPS. Droplet-client proteins have pLLPS < 0.60 but possess droplet-promoting regions, which can induce their partitioning into condensates [96]. Therefore, according to the results of this analysis, P2 and P3 isoforms can potentially serve as droplet-drivers, whereas other isoforms (P1, P4 and P5) most likely represent droplet clients.

On pg. 9 it states, “the MATRX_RABVP protein and shows that a section in the middle of the protein, spanning roughly from residue 50 to residue 175, displays low disorder content and is likely to represent the structural domain of the protein. Curiously, region 115-151, which is essential for the glycoprotein binding includes an IDPR (residues 129- 141), indicating that intrinsic disorder of this region can contribute to its interactability.” These two sentences appear contradictory. The claim is the region defined by residues 50-175 is ordered but then it’s the “disordered” part (129-141) that is important for binding even though the disorder score for this region is ~0.5 and does not make up a majority of the region 115-151.

RESPONSE: Thank you for pointing this out. The corresponding clarifications are added. This section of the manuscript now reads:

Figure 3A represents the disorder profile of the MATRX_RABVP protein and shows that a section in the middle of the protein, spanning roughly from residue 50 to residue 175, displays, on average, low disorder content and is likely to represent the structural domain of the protein, which, however, includes some disordered/flexible regions.

Curiously, region 115-151, which is essential for the glycoprotein binding includes an IDPR (residues 129-141 as per PONDR® VLXT or residues 130-137 as per PONDR® VSL2), indicating that intrinsic disorder (or structural flexibility) of this region can contribute to its interactability.

On pg. 12 it states, “The N-protein also functions to prevent activation of the RIG-I mediated antiviral response [19,20]. The IDPR found in the N-protein may serve to aid in suppressing the antiviral response.” Again there is a lack of logic flow to connect predicted disorder regions to a listed function.

RESPONSE: Thank you for pointing this out. The corresponding clarification was added:

Although the actual information on the molecular mechanisms of this suppression is currently unavailable, it is tempting to hypothesize that the MoRF-containing IDPR found in the N-protein may serve to aid in suppressing the antiviral response.

In the abstract it is stated, “The study suggests that the high levels of intrinsic disorder in the phosphoprotein (P-protein) and nucleoprotein (N-protein) allow them to participate in creation of the Negri bodies and help this virus suppress the antiviral immune response in the host cells”. Yet on pg. 12 it states that the N-protein has a low probability of undergoing LLPS according to FuzDrop. Thus, again there is a lack of connection between the presence of predicted disorder, LLPS, and Negri body formation.

RESPONSE: Thank you for pointing this out. The following clarification was added to the revised manuscript.

Although based on the FuzDrop analysis, the N-protein is expected to have low probability of the spontaneous LLPS (pLLPS = 0.1405) and does not include any DPRs, this RNA-binding protein is invariantly present in Negri bodies (NBs) [110]. It is likely that the involvement of the N-protein in the NB biogenesis is linked to the ability of this protein to bind both viral RNA and P-protein. In fact, no NBs were found when limiting concentrations of one of these proteins were expressed in the model experi-ments [110]. Furthermore, even when the RABV P- and N-proteins were expressed alone (i.e., without viral RNA), they were capable of the formation of the NB-like structures [110]. These observations indicated that in the resulting N-P inclusions, the RABV N-protein was likely bound to the cellular RNAs, forming the N-RNA complex-es similar to viral nucleocapsids [125]. It is known that different regions of the P-protein are utilized in interaction with the N-protein, where the disordered N-terminal domain interacts with the RNA-free N°-protein, whereas the P-protein CTD binds to the RNA-associated N-protein [110,125]. As it was already indicated, the positive patch within the N-terminal region of the P-protein CTD that is actually re-sponsible for the N-protein binding contains flexible/disordered residues, further em-phasizing the potential role of structural disorder/flexibility in the NB biogenesis.

On pg. 14 there is a lack of logic flow for the statement, “The IDPR in the G-protein likely serves to facilitate endocytosis into the cell through interacting with cell recognition molecules.” Also it is unclear why Zhang et al., is cited for this statement given endocytosis is not mentioned in Zhang et al.

RESPONSE: Thank you for pointing this out. This sentence was deleted to avoid confusion and overstatement.

On pg. 14 it states, “The IDPRs likely define the lower fidelity of the polymerase, which results in a high mutation rate and therefore higher flexibility in the ability of the virus to adapt to host defenses [7].” Again there is no logical connection between predicted disorder and function, and the citation (Kouznetzoff et al.) appears to just be about the N protein.

RESPONSE: Thank you for pointing this out. The mentioned citation (Kouznetzoff et al.) was retained by mistake during the conversion of the hand-incorporated references in this manuscript to the EndNote. This issue is fixed now.  We also toned down this statement that now reads:

It is possible that the need for interplay between ordered and disordered features in this protein reflects its purpose to serve as a low fidelity viral polymerase, as intrinsic disorder and structural flexibility likely define the lower fidelity of the polymerase, which results in a high mutation rate and therefore higher flexibility in the ability of the virus to adapt to host defenses [8].

On pg. 17 it states, “This revealed a very high level of disorder in the majority of the proteins from this dataset, with the entire set being characterized by the mean PPIDR of 41.6±20.9 % (as evaluated using the outputs of the PONDR® VSL2 predictor, which is one of the most accurate stand-alone disorder predictors). Supplementary Figures S1, S2, S3, S4, and S5 show that all these proteins contain multiple IDPRs of various length. Many proteins contain multiple MoRFs, and almost all human proteins in this dataset are densely decorated by a multitude of different PTMs. These observations indicate that intrinsic disorder in this these proteins is related to their functionality, playing a role in their binding promiscuity, as evidenced by dense PPI networks centered at these proteins.” There is no reference to whether these observations are statistically significant. This dataset could be directly compared to the full human proteome to determine if the mean PPIDR is significant. And even if the mean PPIDR is greater than the human proteome it does not automatically follow that “intrinsic disorder in [this] these proteins is related to their functionality”.

RESPONSE: Thank you for pointing this out. We conducted suggested disorder analysis for the entire human proteome and added the corresponding data to the revised manuscript.

…indicating that 35.1% of human proteins interacting with RABV are expected to be extremely disordered. These levels of disorder in human RABV interactors are comparable to those observed in the entire human proteome, where out of 20,317 proteins 7,590 (37.3%) and 12,363 proteins (60.8%) are characterized by the PPIDR ≥ 30% and PPIDR ≥ 50%, respectively.

On pg. 17 it states, “In fact, Figure 7A shows there is no a single protein in this dataset that could be classified as mostly ordered, whereas 25 proteins (67.6%) are expected to be mostly disordered (i.e., their PPIDR exceeds 30%).” Why is 30% defined as “mostly disordered”. PONDR defines disorder as anything greater than or equal to 0.5. This is also the line cutoff used in all of the previous disorder predictor plots in this paper. Thus, according to this definition many of the proteins are “mostly” ordered. The reasoning for 30% needs to be addressed and why it is different than what PONDR suggests. For instance, PONDR VSL2 (%) which is defined in the figure caption as the percent of “residues with disorder scores above 0.5” is less than 50% for many proteins which implies that those proteins are mostly ordered. Furthermore, the same symbols and colors are repeated in Figure 7 making it impossible to distinguish between certain proteins.

RESPONSE: We are respectfully disagree with these statements. Used in this study classification is based on the accepted practice. In fact, this is explicitly stated in the Materials and Methods section of the original manuscript (lines 256-261). However, to reemphasize this important point, we added this information to the Results and Discussion section as well.  

This classification is based the accepted in the field practice to group proteins based on their PPIDR values [87], where proteins with PPIDR < 10% are considered as ordered or mostly ordered; proteins with 10% ≤ PPIDR < 30% are considered as moderately disordered; whereas proteins with the PPIDR ≥ 30% are considered as highly disordered [87]

On pg. 17 it states, “Figure 7B shows that quadrant Q1 (bottom left corner) contains 21 proteins that are predicted to be ordered by both predictors; quadrant Q2 (bottom right corner) includes 10 proteins, which are predicted to be ordered/compact by CH-plot and disordered by CDF (i.e., it contains either molten globular proteins, which are compact, but without unique 3D structures, or hybrid proteins containing comparable levels of ordered and disordered residues); quadrant Q3 (top right corner) includes 4 highly disordered (native coils or native pre-molten globules), which are predicted as disordered by both predictors. Finally, one proteins in quadrant Q4 (top left corner) is predicted to be disordered by CH-plot and ordered by CDF-plot. Therefore, 16 human proteins interacting with the RABV proteins are predicted as containing very noticeable levels of disorder” This description does not match the figure. Particularly, the bottom right has the most proteins, but is said to only have 10 proteins. Also where does the number 16 come from when the quadrants contain 21, 10, 4, and 1? Also, if the 21 proteins are predicted to be ordered by both predictors, then this whole plot is in contrast to what was stated about these proteins in Figure 7A, which claimed “no single protein in this dataset could be classified as mostly ordered”.

RESPONSE: Thank you for pointing this out. This was our mistake in describing quadrants. In fact, quadrants Q1, Q2, Q3, and Q4 are located in bottom right corner, bottom left corner, top left corner and top right corner, respectively. We fixed this issue in the revised manuscript accordingly. We also fixed annotations in the legend to Figure 7B and corrected numbers of proteins that are predicted to be ordered by both predictors. The indicated difference between the interpretation of the data shown in Figures 7A and 7B are rooted in the principle differences of the tools utilized for these analyses, where Figure 7A shows outputs of the per-residue predictor, whereas Figure 7B reports data generated by the so-called binary predictors; i.e., tools that classify query proteins as mostly ordered or mostly disordered. Obviously, mostly ordered protein might contain noticeable levels of disordered residues, whereas mostly disordered protein might possess noticeable levels of ordered residues. The corresponding discussion is added to the revise manuscript.

The apparent discrepancies between the data shown in Figures 7A and 7B are rooted in the principle differences of the tools utilized for these analyses, where Figure 7A represents the outputs of the per-residue predictor, whereas Figure 7B reports data generated by the so-called binary predictors; i.e., tools that classify query proteins as mostly ordered or mostly disordered. Obviously, mostly ordered protein might contain noticeable levels of disordered residues, whereas mostly disordered protein might possess noticeable levels of ordered residues.

Minor comments:

On pg. 9 there is no citation for the statement, “Curiously, region 115-151, which is essential for the glycoprotein binding includes an IDPR (residues 129-141), indicating that intrinsic disorder of this region can contribute to its interactability.”

RESPONSE: Thank you for pointing this out. The corresponding information is added to the revied manuscript:

(as per manually asserted information inferred from the sequence similarity and available in the UniProt database, see https://www.uniprot.org/uniprotkb/P08671/entry)

On pg. 10 one of the 3Bs should be 3A in the sentence, “Comparison of the Figures 3B and 3B illustrates a remarkable similarity of the per-residue intrinsic disorder predispositions of the Lagos…”.

RESPONSE: Thank you for pointing this out. This issue was fixed.

On pg. 11 it states, “The N-protein has been shown to form a complex with the P-protein during replication and encapsulate the viral genomic RNA to protect it against nucleases. It binds through the N-terminus of the protein, which is highly intrinsically disordered.” However, there are no citations to support these claims. Also only the first ~10 residues of the N-terminus are predicted to be disordered so is this the region that binds to the Pprotein or a region larger than this? And then in the next sentence it implies the C-terminus is the actual region that interactions with the P-protein. Also the citation that goes with this sentence (Luo et al.) does not show what the sentence claims. This text needs to be cited correctly and clarified.

RESPONSE: Thank you for pointing this out. This section of the manuscript was clarified and the corresponding references were added.

The N-protein, which is the most transcriptionally abundant protein during infection [123], has been shown to encapsulate the viral genomic RNA to protect it against nucleases and form a complex with the P-protein during replication [124]. The RABV P-protein binds to the RNA-free N°-protein through the N-terminus [110,125], which is predicted as highly intrinsically disordered (see Figure 2A), whereas the N-terminal of the P-protein CTD is responsible interaction with the RNA-bound N-protein (see below).

On pg. 12 it states, “The wide curvature of the C terminus of the protein, which is highly intrinsically disordered, makes it able to more tightly cover bound RNA.” What does “wide curvature” mean in terms of region that adopts heterogeneous conformations? The references to curvature from Luo et al. seems to be in terms of the structured domains. The meaning of this statement needs to be clarified.

RESPONSE: Thank you for pointing this out. This sentence was amended to read:

It is likely that the high structural flexibility and pliability of the C-terminal region of the protein, which is predicted as highly intrinsically disordered, makes it able to more tightly cover bound RNA.

On pg. 12 it states, “The C-terminal domain of the G protein is essential for trimer stability”. Please clarify what the “C-terminal domain” refers to. Is it residues 258-505?

RESPONSE: Thank you for pointing this out. The corresponding clarification is added to the revised manuscript.

The C-terminal domain of the G protein (residues 258-505) is essential for trimer stability [9].

On pg. 12 it states, “The most promising IDPR is the C-terminal IDPR (residues 486-524), which corresponds to the intravirion domain of this protein that is engaged in interaction with the matrix protein.” This statement has no citation.

RESPONSE: Thank you for pointing this out. The corresponding clarification is added to the revised manuscript.

The most promising predicted disordered region is the C-terminal IDPR (residues 486-524, see Figure 5A), which corresponds to the intravirion domain of this protein (https://www.uniprot.org/uniprotkb/P08667/entry) that is engaged in interaction with the matrix protein [13].

Line 527 has two Figure 5Bs.

RESPONSE: Thank you for pointing this out. This issue is fixed.

There are several typos on pg. 16 including lines 582 and 605.

RESPONSE: Thank you for pointing this out. These issues were fixed.

It is not stated how “mostly disordered” is calculated for Figure 9 and there is no color bar. If the authors are using 30%, then this value needs to be explained because 30% does not generally imply the value is high.

RESPONSE: Thank you for pointing this out. The corresponding clarifications were already added to the revised text, while discussing disorder classification used in this study.

Round 2

Reviewer 2 Report

Please see below for my additional comments on the manuscript.

The authors state, “Negri bodies, which are inclusions in the host cytoplasm used for viral replication, are formed from the interaction of these two intrinsically disordered regions and the C-terminal domain of P with the intrinsically disordered regions of the N-protein [99].” However, the paper cited states, “for both RABV and MeV P, it has been shown that the oligomerization domain, the second intrinsically disordered domain (IDD2/PLoop) and the C-terminal domain (PCTD/XD) are required for IB formation whereas the N-terminal part and the first intrinsically disordered domain are dispensable.” Thus, this statement must be clarified.

All section titles should read something to the regards of “Disorder of the X-protein and its suggested functional consequences” instead of “Functional disorder of the X-protein”, since the study is not providing direct evidence for functional relevance of the disordered regions.

Line 826 “confirmations” should be “conformations”.

Line 1011 “where” should be “were”.

The authors state, “This revealed a very high level of disorder in the majority of the proteins from this dataset, with the entire set being characterized by the mean PPIDR of 41.6±20.9 % (as evaluated using the outputs of the PONDR® VSL2 predictor, which is one of the most accurate stand-alone disorder predictors).” There should be a citation.

The authors state, “These levels of disorder in human RABV interactors are comparable to those observed in the entire human proteome, where 1082 out of 20,317 proteins 7,590 (37.3%) and 12,363 proteins (60.8%) are characterized by the PPIDR≥ 30% and PPIDR ≥ 50%, respectively.” The order of this sentence does not match the “respectively” order.

Figure 7 should still be remade such that the symbols aren’t just repeating.

The authors state, “This diagram clearly shows that most of the proteins (viral and human) in this network are “red” (highly disordered), and there are no “blue” (mostly ordered) proteins, suggesting importance of intrinsic disorder for RABV infection.” Since a “blue” option is stated in this sentence, please provide a color bar or color key for Figure 9.

The authors state, “All Rabies Lyssavirus PV proteins contain IDPRs, most of which aid in the flexibility of the virus and its ability to evade host antiviral defenses.” Please add a clarifier and / or tone down this sentence since it is not proven that “most of which aid…”.

Author Response

Please see below for my additional comments on the manuscript.

The authors state, “Negri bodies, which are inclusions in the host cytoplasm used for viral replication, are formed from the interaction of these two intrinsically disordered regions and the C-terminal domain of P with the intrinsically disordered regions of the N-protein [99].” However, the paper cited states, “for both RABV and MeV P, it has been shown that the oligomerization domain, the second intrinsically disordered domain (IDD2/PLoop) and the C-terminal domain (PCTD/XD) are required for IB formation whereas the N-terminal part and the first intrinsically disordered domain are dispensable.” Thus, this statement must be clarified.

RESPONSE: Thank you for pointing this out. The corresponding changes and clarifications were introduced.

All section titles should read something to the regards of “Disorder of the X-protein and its suggested functional consequences” instead of “Functional disorder of the X-protein”, since the study is not providing direct evidence for functional relevance of the disordered regions.

RESPONSE: Changed as recommended

Line 826 “confirmations” should be “conformations”.

RESPONSE: Corrected

Line 1011 “where” should be “were”.

RESPONSE: Corrected

The authors state, “This revealed a very high level of disorder in the majority of the proteins from this dataset, with the entire set being characterized by the mean PPIDR of 41.6±20.9 % (as evaluated using the outputs of the PONDR® VSL2 predictor, which is one of the most accurate stand-alone disorder predictors).” There should be a citation.

RESPONSE: Requested reference is added

The authors state, “These levels of disorder in human RABV interactors are comparable to those observed in the entire human proteome, where 1082 out of 20,317 proteins 7,590 (37.3%) and 12,363 proteins (60.8%) are characterized by the PPIDR≥ 30% and PPIDR ≥ 50%, respectively.” The order of this sentence does not match the “respectively” order.

RESPONSE: Thank you for pointing this out. The corresponding changes were introduced.

Figure 7 should still be remade such that the symbols aren’t just repeating.

RESPONSE: Thank you for pointing this out. Figure 7 was changed accordingly.

The authors state, “This diagram clearly shows that most of the proteins (viral and human) in this network are “red” (highly disordered), and there are no “blue” (mostly ordered) proteins, suggesting importance of intrinsic disorder for RABV infection.” Since a “blue” option is stated in this sentence, please provide a color bar or color key for Figure 9.

RESPONSE: Thank you for pointing this out. The corresponding clarification was added to the legend of Figure 9.

The authors state, “All Rabies Lyssavirus PV proteins contain IDPRs, most of which aid in the flexibility of the virus and its ability to evade host antiviral defenses.” Please add a clarifier and / or tone down this sentence since it is not proven that “most of which aid…”.

RESPONSE: Thank you for pointing this out. This sentence is changed to read: All Rabies Lyssavirus PV proteins contain IDPRs, most of which are expected to aid in the flexibility of the virus and its ability to evade host antiviral defenses.